# MI-Pruner: Crossmodal Mutual Information-guided Token Pruner for Efficient MLLMs

## Abstract

In multimodal large language models (MLLMs), visual tokens are characterized by their high volume and inherent sparsity compared with the text counterparts. To achieve efficient inference with controllable token budgets, training-free token pruning techniques emerge for their versatility and near-zero cost. Current methods typically measure token importance based on attention salience in the visual encoder or the LLM decoder, then preserve visual tokens with high attention scores while pruning others. However, attention salience is often biased by sink tokens and positional bias. These salience-based methods require extracting attention maps, which introduces implementation complexity and memory overhead, while inadequately accounting for the diversity of selected tokens. In this paper, we pursue a sound and surgical approach, called **MI-Pruner**, which detours attention collection and instead estimates Mutual Information (MI) based relevance in the projection space. This allows an explicit measure of feature-level dependency with information-theoretic motivation to identify the most informative tokens. Without reliance on internal attention maps or architectural modifications, **MI-Pruner** can be seamlessly applied to off-the-shelf MLLMs for inference acceleration. Extensive experiments on LLaVA1.5, Qwen-series and Video-LLaVA demonstrate that our approach achieves a favorable performance-efficiency trade-off across diverse image and video understanding benchmarks.

## 1 Introduction

> *"Tell the whole story from just one clue." — A principle*

Modern multimodal models have rapidly grown in scales and capabilities (Bai et al., 2025; Chen et al., 2024c; Liu et al., 2024b), which comes with a substantial computational cost. Notably, the quadratic complexity of the attention mechanism makes inference increasingly expensive as the number of tokens grows when processing high-resolution visual inputs (Li et al., 2024a). To address it, token pruning has emerged as a critical technique for efficient inference, aiming to retain only the most informative tokens while discarding redundant ones (Ye et al., 2025; Cao et al., 2024). By reducing sequence length in the forward pass, the pruning not only alleviates computation and memory consumption, but also enables scalable deployment of large multimodal models in latency- and resource-constrained scenarios. However, identifying informative tokens remains challenging (Fan et al., 2025). And erroneous pruning of evidence tokens leads to LLM hallucination, where responses are generated merely by prior knowledge in LLMs rather than input evidence (Bai et al., 2024).

Salience-based pruning emerges by collecting attention maps from vision encoders or LLMs. Motivated by studies on multimodal information flow (Zhang et al., 2025g;e), some work (Chen et al., 2024a; Zhang et al., 2025f) considers visual dilution after LLM shallow layers and selects tokens by analyzing the cross-attention matrix in LLM decoders. Yet, the entangled multimodal semantics in LLMs contain inherent noise (Zhang et al., 2025c; Cai et al., 2025) to distort the importance score. As a result, recent approaches (Zhang et al., 2025c; Yang et al., 2025a; Cai et al., 2025) instead turn to visual cues for token pruning, *e.g.*, self-attention scores extracted in the vision encoder. From the perspective of information theory, the attention mechanism employs learnable QKV matrices to capture token-wise dependencies, and the attention scores work as an

approximation to Pointwise Mutual Information (PMI). Eventually, the attention module learns an implicit distribution that maximizes the Mutual Information between queries and values. Despite the progress, there exist several shortcomings in attention-based pruning. ***(i) Heuristic metrics with miscalibrated attention:*** The raw attention scores don't fully equal semantic importance due to the positional bias (Wen et al., 2025a), attention sinks (Xiao et al., 2024; Yang et al., 2025c) and multi-head diversity (Zhang et al., 2025e; Kang et al., 2025a). As observed, attention maps have outliers incongruous with local semantics (Gu et al., 2025), and the multi-head average obliterates the head-specific roles (Yang et al., 2025c; Kang et al., 2025a). Therefore, a necessary calibration step is missing in previous salience-based methods (Darcet et al., 2024; Laurençon et al., 2024; Yan et al., 2025). ***(ii) Imbalanced diversity:*** Prioritizing high attention scores tends to preserve salient but overly similar tokens (Zou et al., 2025). The imbalanced concern of salience and diversity leads to a substantial performance drop under high pruning ratios, since the visual importance is not an isolated metric but depends on textual queries and visual contexts. ***(iii) Implementation bottlenecks:*** The inner-LLM operations (Chen et al., 2024a; Zhang et al., 2025f) are incompatible with the latest attention cores like FlashAttention (Dao, 2024), while collecting attention maps from visual encoders increases memory overhead and overlooks the multimodal nature.

Unlike the accumulated salience from attention maps (Yang et al., 2025a; Zhang et al., 2025c), we propose a surgical and training-free method derived from Mutual Information and conduct greedy search using our MI-based relevance. In the projection space, we construct temperature-scaled similarity matrices from multimodal features, then apply a softmax operation to formulate conditional and marginal probabilities. These probabilities are subsequently used to compute Pointwise Mutual Information and aggregated for our MI-based relevance scores. Finally, we conduct greedy selection by crossmodal and intra-modal MI-based relevance scores. Different from the attention mechanism, we derive probabilities and token-wise dependency by revisiting information theory. As a result, our **MI-Pruner** circumvents extra learnable QKV matrices and the collection of attention maps from vision encoders (Yang et al., 2025a) or LLMs (Chen et al., 2024a). Grounded in information quantities (Iyer et al., 2021; Spadaro et al., 2023; Gao et al., 2020; Potkins et al., 2024), our MI-based scoring function effectively identifies irreplaceable and query-relevant vision tokens. As a surgical and model-agnostic token reduction approach, **MI-Pruner** supports non-intrusive pruning and can be easily applied to a wide range of MLLMs following the Enc-MLP-Dec paradigm. Experiments across various benchmarks demonstrate that our method achieves highly competitive performance and significantly accelerates the inference. Moreover, we present **MI-Attention**, a unified approach integrating our MI-based scoring and attention salience for flexible usage.

In summary, our **contributions** are:

1. ~~We propose **MI-Pruner**, a plug-in visual token pruning method without relying on internal attention maps. By maximizing an MI-based relevance score in the projection space, **MI-Pruner** efficiently captures visual cues aligned with the textual query.~~ **We propose MI-Pruner, an attention-free and query-aware visual token pruning method, which is achieved by PMI estimation and greedy search in the projection space.**

2. As a model-agnostic module, **MI-Pruner** can be seamlessly integrated into various off-the-shelf MLLMs with controllable budgets. Moreover, our approach can be integrated into salience-based methods[1] to further refine the token selection process.

3. Empirical results on LLaVA1.5 and Qwen-series indicate our effectiveness and efficiency, which reveal that attention scores are not the only measure of semantic importance.

## 2 Related Work

**Efficient MLLMs.** To reduce the computational costs and inference latency, a surge of research effort has been dedicated to efficient MLLMs. For instance, the *Mixture-of-Depth* (MoD) series learn to assess the token importance in LLM, where p-MoD (Zhang et al., 2025b) designs a progressive ratio decay from

---

[1]We replace similarity measures (in the 2-round) of previous attention salience-based methods (Yang et al., 2025a; Zhang et al., 2025c) with our MI-based scores, which is denoted as **MI-Attention** in the following sections.

shallow to deep layers. In addition, *speculative decoding* leverages a draft model to generate a bunch of tokens and a large model to do one-time validation. Its application ranges from LLMs (Li et al., 2024b;c; 2025b) to MLLMs (Ji et al., 2025). Moreover, recent research (Yang et al., 2025b; Liao et al., 2026) observed that low-resolution visual inputs with reduced tokens are sufficient for most scenarios, and explored adaptive resolutions by *reinforcement learning* (Shabtay et al., 2026). Beyond these techniques, token pruning stands out for its training-free nature and controllable budgets. In the following, we analyze two mainstream token pruning methods according to attention salience and subset coverage.

**Salience-based pruning.** Many works (Chen et al., 2024a; Fan et al., 2025; Lin et al., 2025) observed that image tokens receive less attention in LLMs' deep layers, which boosts attention salience-based visual compression in LLMs. As a pioneer, FastV (Chen et al., 2024a) discards non-salient [`vis`] tokens guided by LLM's cross-attention scores. The following work, SparseVLM (Zhang et al., 2025f) selects a few [`text`] tokens as raters, then calculates the rank of attention matrices for dynamically pruning in LLMs. Another branch argues that purely visual cues are sufficient to indicate the informative patches. VisionZip (Yang et al., 2025a) first selects image tokens with high [`CLS`] attention in the last layer of the vision encoder (round-1) and then merges the remaining tokens by semantic similarity (round-2) to compress the LLM inputs. Similarly, VisPruner (Zhang et al., 2025c) progressively removes duplicates by similarity in round-2. Despite working in various stages, attention salience-based methods often suffer from semantic miscalibration, diversity imbalance and implementation bottlenecks.

**Subset-based pruning.** A bunch of work stops chasing the highlight tokens from attention maps but turns to study the problem of subset coverage, *e.g.* similarity or conditional diversity measures. DART (Wen et al., 2025b) introduces pivot tokens in LLMs to emphasize the (dis-)similarity over importance, while ApET (Ma et al., 2026) turns to measure how well the selected set covers a new token. Divprune (Alvar et al., 2025) and CDPruner (Zhang et al., 2025d) leverage Min-Max diversity and conditional similarity to score visual tokens, and HoloV (Zou et al., 2025) considers both attention and diversity for a balanced metric. Akin to ours, MMToK (Dong et al., 2026) constructs an energy-based similarity matrix, yet their analysis ends at empirical coverage. As a classic information processing tool, Mutual Information was also introduced into token pruning. For instance, AutoPrune (Wang et al., 2025a) interprets attention scores as probabilities in their MI scores, assuming uniform text tokens. Under conditional Gaussian distributions, TrimTokenator (Zhang et al., 2025a) adopts the L2-norm as MI proxy for visual pruning. In comparison, **MI-Pruner** conduct softmax over projection-space features to derive probabilities used in MI-based relevance scores, which avoids latency from collecting attention maps and the miscalibration issue (Huang et al., 2026). Our contribution is not the first MI-based token pruning method, but an efficient PMI-based token compression framework, which is attention-free and easy to integrate.

## 3 Preliminaries

### 3.1 MLLM Architecture

The general MLLM architectures (Liu et al., 2024a; Bai et al., 2025) follow an Enc-MLP-Dec paradigm, including a vision encoder $\mathbf{E_v}$ with an MLP projection module $g$, a tokenizer $\mathbf{E_T}$, and an LLM decoder. Given an image $I$ and a prompt $P$ (system instructions omitted), multimodal features $\mathcal{V}, \mathcal{T}$ are extracted as:

$$\mathcal{V} := \{\mathbf{v}_i\}_{i=1}^{N_V} = g(\mathbf{E_V}(I)) \in \mathbb{R}^{N_V \times d}, \quad \mathbf{v}_i \in \mathbb{R}^d, \tag{1}$$

$$\mathcal{T} := \{\mathbf{t}_j\}_{j=1}^{N_T} = \mathbf{E_T}(P) \in \mathbb{R}^{N_T \times d}, \quad \mathbf{t}_j \in \mathbb{R}^d. \tag{2}$$

Then, $N_V$ image features and $N_T$ text features are concatenated as inputs for LLM decoding:

$$\mathcal{X} := \mathcal{V} \oplus \mathcal{T} \in \mathbb{R}^{(N_V + N_T) \times d}. \tag{3}$$

### 3.2 Visual Attention for Pruning

To assess token importance, methods based on visual cues (Zhang et al., 2025c; Yang et al., 2025a) typically employ a two-step pruning strategy: (i) aggregate attention scores from [`CLS`] or other image tokens from $\mathcal{V}$

as $\mathrm{Attn}_i$, to determine the initial pruning budget; (ii) recycle from remaining tokens $\mathcal{V}_{\mathrm{re}}$ by their similarity $S_i$. Let cls denote the [CLS] index in the attention matrix $\mathbf{A} \in \mathbb{R}^{N_V \times N_V}$, and let $\tilde{\mathbf{v}}_i \in \mathbb{R}^d$ denote the normalized visual features before projection. The two-step importance measures are formulated as:

$$\mathrm{Attn}_i = \mathbf{A}_{\mathrm{cls},i} \text{ or } \sum_{j \neq i} \mathbf{A}_{ji}, \tag{4}$$

$$S_i = \sum_{j \neq i, \tilde{\mathbf{v}}_j \in \mathcal{V}_{\mathrm{re}}} (\tilde{\mathbf{v}}_i^\top \tilde{\mathbf{v}}_j), \tag{5}$$

where $\cdot^\top$ denotes the transpose operator. Our approach can be integrated into attention salience-based methods like VisPruner and VisionZip by replacing Eqn. (5) with our MI-based scoring function, denoted as **MI-Attention** in the following.

### 3.3 Information Theory

Information theory provides a rigorous mathematical framework for quantifying information transmission and representation, in which Mutual Information serves as a natural measure of how much information is shared between variables. Applied to MLLMs, MI is able to characterize the statistical dependence among tokens, offering a principled objective for identifying important and irreplaceable tokens.

**Definition 1** (Mutual Information (Kraskov et al., 2004; Gray, 2011)). *The Mutual Information* $\mathrm{MI}(X;Y)$ *measures the gap between entropy* $H(X)$ *and conditional entropy* $H(X|Y)$ *(see App. A.1 for details), which quantifies the uncertainty reduction of a random variable (r.v.)* $X$ *given the knowledge of another r.v.* $Y$:

$$\mathrm{MI}(X;Y) = H(X) - H(X \mid Y) \tag{6}$$

$$= \sum_{x \in \mathcal{X}, y \in \mathcal{Y}} p(x,y) \underbrace{\log \frac{p(x \mid y)}{p(x)}}_{\mathrm{PMI}(x;y)}. \tag{7}$$

$\mathrm{MI}(X;Y)$ can be interpreted as the weighted average of Pointwise Mutual Information $\mathrm{PMI}(x,y)$ under the joint distribution $p(x,y)$. Replacing $x, y$ with embeddings $\mathbf{v}_i, \mathbf{t}_j$ establishes a link to multimodal token relevance. In general, maximizing MI is computationally intractable. However, under the assumption[2] of conditional independence (the components of $X$ are independent given $Y$), it becomes a cardinality-constrained monotone submodular maximization problem (Krause et al., 2008). This is a well-studied problem with a known approximation guarantee of $1 - 1/e$. In the following, we introduce the marginal gain and diminishing returns of submodular functions.

**Definition 2** (Marginal Gain (Fujishige, 2005)). *Let* $\mathcal{X}$ *be a ground set, and let* $f : 2^{\mathcal{X}} \to \mathbb{R}$ *be a set function. The marginal gain of adding an element* $i \in \mathcal{X} \setminus \mathcal{A}$ *to a selected set* $\mathcal{A} \subseteq \mathcal{X}$ *is defined as:*

$$\Delta f(i|\mathcal{A}) = f(\mathcal{A} \cup \{i\}) - f(\mathcal{A}). \tag{8}$$

**Definition 3** (Submodularity and Diminishing Returns (Fujishige, 2005)). *A function* $f$ *is submodular if and only if for all* $\mathcal{A} \subseteq \mathcal{B} \subseteq \mathcal{X}$ *and* $i \in \mathcal{X} \setminus \mathcal{B}$:

$$\Delta f(i|\mathcal{A}) \geq \Delta f(i|\mathcal{B}). \tag{9}$$

The ***monotonicity*** motivates our MI-guided visual selection. Let $\mathcal{V}_{\mathrm{S}} \subseteq \mathcal{V}$ be a selected set of a ground set of variables, and let $\mathcal{T}$ be a targeted set of variables, we define Mutual Information as a set function:

$$f(\mathcal{V}_{\mathrm{S}}) = \mathrm{MI}(\mathcal{V}_{\mathrm{S}}; \mathcal{T}). \tag{10}$$

---

[2]This assumption is necessary to make MI tractable, and has been implicitly adopted in prior studies employing greedy search (see Related Work).

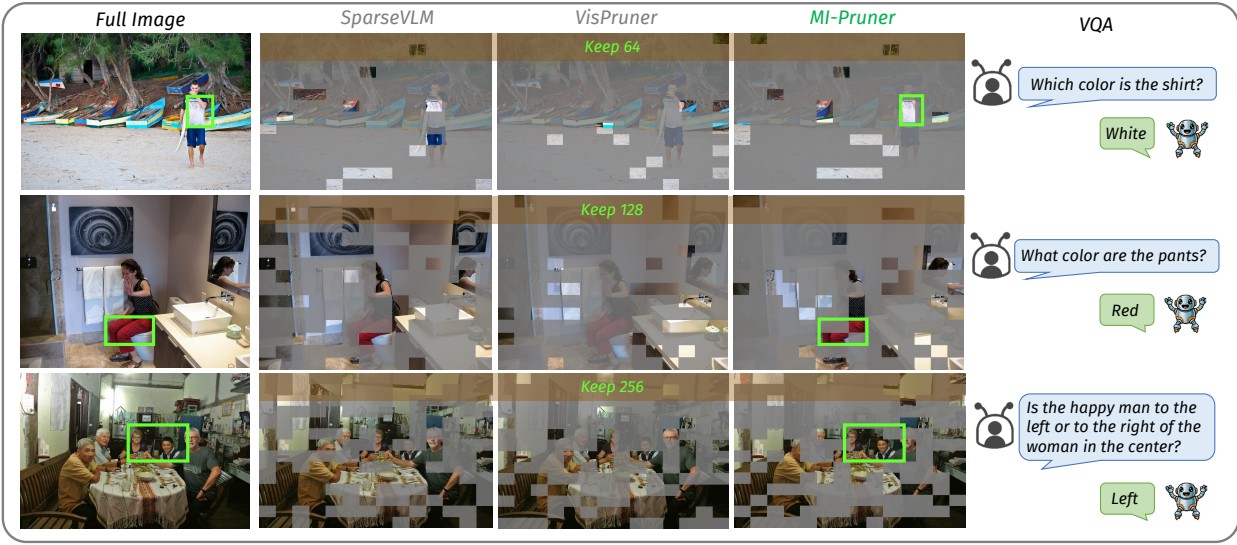

Figure 1: **Pruning visualization on LLaVA1.5-7B with different budgets.** Our **MI-Pruner** consistently identifies and preserves the queried regions, whereas other methods partially miss relevant information. Tokens from top to bottom: 64, 128, 256.

Following Eqn. (8), the marginal gain of adding a new token $\mathbf{v}_i$ into $\mathcal{V}_S$ is expressed as:

$$\Delta f(\mathbf{v}_i \mid \mathcal{V}_S) = \mathrm{MI}(\mathcal{V}_S \cup \{\mathbf{v}_i\}; \mathcal{T}) - \mathrm{MI}(\mathcal{V}_S; \mathcal{T}) \tag{11}$$
$$= \mathrm{MI}(\mathbf{v}_i; \mathcal{T} \mid \mathcal{V}_S). \tag{12}$$

The definition of conditional MI is given in App. A.1 (Def. 6), and the proof from Eqn. (11) to (12) can be found in App. A.2.

The cardinality-constrained monotone submodular maximization problem is typically addressed by a greedy search strategy, which selects elements that maximize the marginal gain per step (Nemhauser et al., 1978):

$$\mathbf{v}^\star = \arg \max_{\mathbf{v}_i \in \mathcal{V} \setminus \mathcal{V}_S} \Delta f(\mathbf{v}_i | \mathcal{V}_S). \tag{13}$$

When selecting $N_{\mathrm{keep}}$ from $N_V$ elements, the greedy search holds a complexity of $\mathcal{O}(N_V N_{\mathrm{keep}})$. In practice, greedy search is a common optimization strategy in token pruning (Alvar et al., 2025; Zhang et al., 2025d). Building upon it, we develop a selection mechanism with even lower complexity.

## 4 Methods

### 4.1 Overview

Visual information is inherently sparser than text (Tang et al., 2025; Chen et al., 2024b), while the image tokens usually consume 10∼100× tokens than their textual counterparts. As illustrated in Fig. 1, repetitive regions are ubiquitous in visual data, such as stretches of beach, flooring and walls. The relative region regarding a concrete query takes even less portion, *e.g.*, a shirt. To preserve query-relevant tokens while pruning visually redundant ones, we introduce Mutual Information-based scoring functions to model cross- and intra-modal relevance in the projection space, where visual and textual features are aligned but have not yet undergone multimodal interaction.

Our pruning consists of 4 steps. Firstly, we construct similarity matrices from normalized projection-space features. Then, we derive conditional probabilities by applying a softmax operation over the similarity matrix, and formulate marginal probabilities by assuming a uniform prior as Wang et al. (2025a). The third

step is to calculate Pointwise Mutual Information from the derived probabilities. Finally, the PMI measures are aggregated into MI-based relevance scores used for greedy selection. See Algorithm 1 for our pipeline.

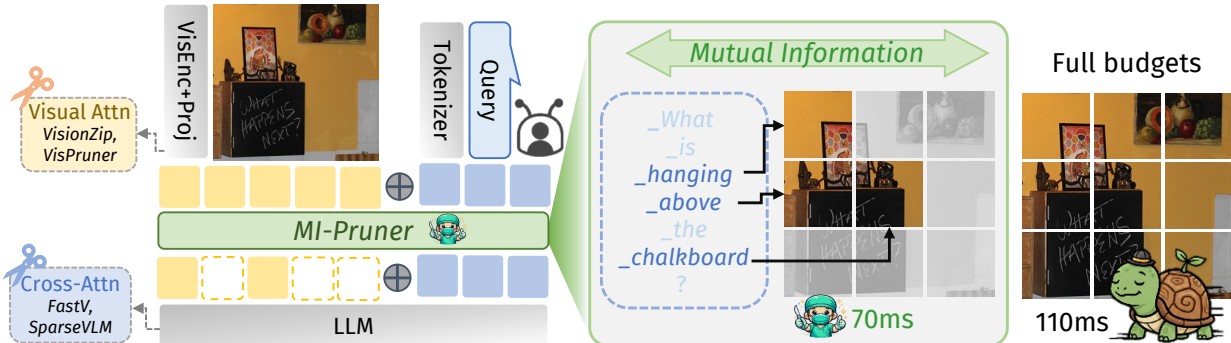

Figure 2: **Overview.** Previous methods prune tokens by collecting attention maps from the vision encoder or LLM decoder. Motivated by Mutual Information, our **MI-Pruner** estimates cross- and intra-modal dependency in the projection space, achieving optimal performance with minimal latency.

## 4.2 MI-Pruner

As shown in Fig. 2, our pruning starts from visual features $\mathcal{V} = \{\mathbf{v}_i\}_{i=1}^{N_V}$, and textual features $\mathcal{T} = \{\mathbf{t}_j\}_{j=1}^{N_T}$, where $\mathbf{v}_i, \mathbf{t}_j \in \mathbb{R}^d$. The objective is to prune visual patches that are weakly correlated with the textual semantics, yet highly redundant among selected patches. To save space, we adopt a unified notation: $\mathrm{s} \in \{\mathrm{cross}, \mathrm{self}\}, \mathbf{x} \in \{\mathbf{t}, \mathbf{v}\}, N \in \{N_T, N_V\}$. **Moreover, we bring some assumptions for computational benefits and better generalization: (i) conditional independence, (ii) uniform prior probability. More discussions can be found in App. B.5.**

**Similarity matrices.** We first conduct normalization to $\mathcal{V}$ and $\mathcal{T}$, yielding normalized features $\tilde{\mathcal{V}}$ and $\tilde{\mathcal{T}}$. Based on Boltzmann distributions, the similarity matrices for vision-text ($\boldsymbol{\rho}_{ij}^{\mathrm{cross}}$) and vision-vision ($\boldsymbol{\rho}_{ij}^{\mathrm{self}}$) pairs are computed as:

$$\boldsymbol{\rho}_{ij}^{\mathrm{cross}} = \frac{\tilde{\mathbf{v}}_i^\top \tilde{\mathbf{t}}_j}{\tau}, \quad \boldsymbol{\rho}_{ij}^{\mathrm{self}} = \frac{\tilde{\mathbf{v}}_i^\top \tilde{\mathbf{v}}_j}{\tau}, \tag{14}$$

where the temperature $\tau$ controls the sharpness of the distribution. For simplicity, we adopt a shared $\tau$ for both matrices, though it's not mandatory. Treating these similarity scores $\{\boldsymbol{\rho}_{ij}^{\mathrm{s}}\}_{j=1}^{N}$ as logits, we define the energy-based crossmodal and internal conditional distribution.

**Conditional distribution.** Applying a softmax operation along the second dimension, we obtain conditional probabilities $p(\tilde{\mathbf{t}}_j \mid \tilde{\mathbf{v}}_i)$, $p(\tilde{\mathbf{v}}_j \mid \tilde{\mathbf{v}}_i)$ in a unified formula:

$$p(\tilde{\mathbf{x}}_j \mid \tilde{\mathbf{v}}_i) = \mathrm{softmax}_j(\boldsymbol{\rho}_{ij}^{\mathrm{s}}) = \frac{\exp(\boldsymbol{\rho}_{ij}^{\mathrm{s}})}{\sum_{k=1}^{N} \exp(\boldsymbol{\rho}_{ik}^{\mathrm{s}})}. \tag{15}$$

It quantifies the semantic alignment between $(\tilde{\mathbf{v}}_i, \tilde{\mathbf{t}}_j)$ and the internal diversity among $(\tilde{\mathbf{v}}_i, \tilde{\mathbf{v}}_j)$, as illustrated by the row operation in Fig. 3.

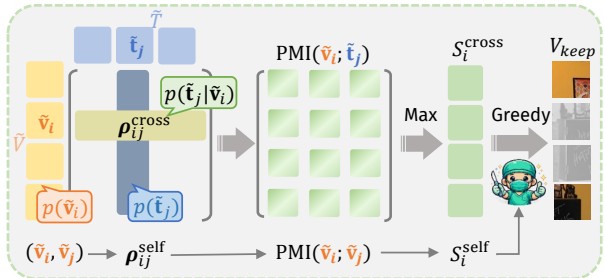

Figure 3: **A toy model of MI-based pruning.** All [vis] tokens are flattened for illustration.

**Marginal distribution.** For the prior probability, we assume $p(\tilde{\mathbf{v}}_j) = \frac{1}{N_V}$, where visual embeddings occur with equal probabilities to avoid spatial bias. According to the law of total probability, the marginal distribution of text embeddings is expressed as:

$$p(\tilde{\mathbf{t}}_j) = \frac{1}{N_V} \sum_{i=1}^{N_V} p(\tilde{\mathbf{t}}_j \mid \tilde{\mathbf{v}}_i), \tag{16}$$

as illustrated by the column operation in Fig. 3. More details can be found in App. A.4.

**Pointwise Mutual Information.** Pointwise Mutual Information measures the token-wise relevance. We adopt a unified notation $\tilde{\mathbf{x}}_j \in \{\tilde{\mathbf{t}}_j, \tilde{\mathbf{v}}_j\}$ to denote pairwise PMI:

$$\text{PMI}(\tilde{\mathbf{v}}_i; \tilde{\mathbf{x}}_j) = \log \frac{p(\tilde{\mathbf{x}}_j \mid \tilde{\mathbf{v}}_i)}{p(\tilde{\mathbf{x}}_j)}. \tag{17}$$

To elaborate on that, $\text{PMI}(\tilde{\mathbf{v}}_i; \tilde{\mathbf{t}}_j)$ quantifies the crossmodal semantic alignment, while $\text{PMI}(\tilde{\mathbf{v}}_i; \tilde{\mathbf{v}}_j)$ captures the intra-modal redundancy. The pairwise PMI scores are aggregated for our MI-based relevance.

**Mutual Information-based relevance.** Eqn. (7) transfers pairwise PMI into MI by a weighted average, and justifies the following greedy search. In token pruning, the weighted average works as *global aggregation* over all text tokens or selected image tokens, which brings the consensus MI-based relevance $S_i^{\text{s}}$ ($\text{s} \in \{\text{cross}, \text{self}\}$):

$$S_i^{\text{s}} = \sum_{j=1}^{N_X} \underbrace{p(\tilde{\mathbf{v}}_i; \tilde{\mathbf{x}}_j)}_{p(\tilde{\mathbf{x}}_j|\tilde{\mathbf{v}}_i)} \cdot \text{PMI}(\tilde{\mathbf{v}}_i; \tilde{\mathbf{x}}_j). \tag{18}$$

Yet, the consensus over many text tokens or image patches brings smoothing effects with diluted significance. We instead implement a *maximal aggregation* strategy pointing for keywords and discriminative patches.

$$S_i^{\text{cross}} = \max_{\tilde{\mathbf{t}}_j \in \tilde{\mathcal{T}}} \text{PMI}(\tilde{\mathbf{v}}_i; \tilde{\mathbf{t}}_j), \tag{19}$$

$$S_i^{\text{self}} = \max_{\tilde{\mathbf{v}}_j \in \tilde{\mathcal{V}}_{\text{S}}} \text{PMI}(\tilde{\mathbf{v}}_i; \tilde{\mathbf{v}}_j). \tag{20}$$

And we define an MI-based scoring function $S_i$ as the marginal gain of adding $\mathbf{v}_i$ based on Eqn. (19-20):

$$S_i = \lambda S_i^{\text{cross}} - (1 - \lambda) \cdot S_i^{\text{self}} \tag{21}$$

$$= \lambda \cdot \underbrace{\max_{\tilde{\mathbf{t}}_j \in \tilde{\mathcal{T}}} \text{PMI}(\tilde{\mathbf{v}}_i; \tilde{\mathbf{t}}_j)}_{\text{relevance} \to \text{sorting}} - (1 - \lambda) \underbrace{\max_{\tilde{\mathbf{v}}_j \in \tilde{\mathcal{V}}_{\text{S}}} \text{PMI}(\tilde{\mathbf{v}}_i; \tilde{\mathbf{v}}_j)}_{\text{redundancy} \to \text{greedy}}. \tag{22}$$

Here, $\lambda=1$ considers only crossmodal relevance towards user prompts, and $\lambda=0$ measures only internal redundancy. Compared with consensus, this prominent relevance captures the strongest crossmodal relevance ("keywords") and intra-modal redundancy ("discriminative patches"). We justify that two aggregation strategies are towards the same optimization direction in App. A.3, and present empirical comparisons in Sec. 5.5.

**Greedy search.** Finally, we conduct a greedy search following Eqn. (13), to find $\mathbf{v}^{\star}$ that maximizes marginal gain per step:

$$\mathbf{v}^{\star} = \arg \max_{\mathbf{v}_i \in \mathcal{V} \setminus \mathcal{V}_{\text{S}}} S_i. \tag{23}$$

## 4.3 Efficient Inference

Given the full token budget described in Sec. 3.1, **MI-Pruner** prunes vision tokens $\mathcal{V} \in \mathbb{R}^{N_V \times d}$ to a smaller set $\mathcal{V}_{\text{keep}} \in \mathbb{R}^{N_{\text{keep}} \times d}$. Consequently, the input of LLMs in Eqn. (3) is updated to $\mathcal{X} \in \mathbb{R}^{(N_{\text{keep}} + N_T) \times d}$:

$$\mathcal{X} = \mathcal{V}_{\text{keep}} \oplus \mathcal{T} \in \mathbb{R}^{(N_{\text{keep}} + N_T) \times d}, \ N_{\text{keep}} < N_V. \tag{24}$$

Since we don't require attention collection or intervention of LLM, our MI-based calculation is very efficient with minimal latency. Meanwhile, the subset selection problem becomes *modular* instead of just submodular in our derivation. Following the scoring-and-selection paradigm in general pruning methods (Wang et al., 2025a; Zhang et al., 2025d), we analyze the computational complexity in these two parts.

---

**Algorithm 1 MI-Pruner**

---

1: **Input:** Visual and textual embeddings $\mathcal{V} = \{\mathbf{v}_i\}_{i=1}^{N_V}$, $\mathcal{T} = \{\mathbf{t}_j\}_{j=1}^{N_T}$, temperature $\tau$, budget $N_{\text{keep}}$

2: **Output:** The selected visual subset $\mathcal{V}_{\text{keep}}$

3: *// Similarity matrices*

4: $\tilde{\mathbf{v}}_i \leftarrow \mathbf{v}_i/\|\mathbf{v}_i\|, \tilde{\mathbf{t}}_j \leftarrow \mathbf{t}_j/\|\mathbf{t}_j\|$

5: $\boldsymbol{\rho}_{ij}^{\text{cross}} \leftarrow \tilde{\mathbf{v}}_i^\top \tilde{\mathbf{t}}_j/\tau, \boldsymbol{\rho}_{ij}^{\text{self}} \leftarrow \tilde{\mathbf{v}}_i^\top \tilde{\mathbf{v}}_j/\tau$

6: *// Derive distributions*

$\quad$ $p(\tilde{\mathbf{t}}_j|\tilde{\mathbf{v}}_i) \leftarrow \text{softmax}_j(\boldsymbol{\rho}_{ij}^{\text{cross}}), \; p(\tilde{\mathbf{v}}_j|\tilde{\mathbf{v}}_i) \leftarrow \text{softmax}_j(\boldsymbol{\rho}_{ij}^{\text{self}})$

7: $p(\tilde{\mathbf{v}}_j) \leftarrow \frac{1}{N_V}, p(\tilde{\mathbf{t}}_j) \leftarrow \frac{1}{N_V}\sum_{i=1}^{N_V} p(\tilde{\mathbf{t}}_j \mid \tilde{\mathbf{v}}_i)$

8: *// Pointwise Mutual Information (PMI)*

$\quad$ $\text{PMI}(\tilde{\mathbf{v}}_i; \tilde{\mathbf{t}}_j) \leftarrow \log \frac{p(\tilde{\mathbf{t}}_j|\tilde{\mathbf{v}}_i)}{p(\tilde{\mathbf{t}}_j)}, \text{PMI}(\tilde{\mathbf{v}}_i; \tilde{\mathbf{v}}_j) \leftarrow \log \frac{p(\tilde{\mathbf{v}}_j|\tilde{\mathbf{v}}_i)}{p(\tilde{\mathbf{v}}_j)}$

9: *// Greedy selection*

10: Initialize $\mathcal{V}_{\text{S}} = \emptyset$

11: **while** $|\mathcal{V}_{\text{S}}| < N_{\text{keep}}$ **do**

12: $\quad$ **for** each $\mathbf{v}_i \in \mathcal{V} \setminus \mathcal{V}_{\text{S}}$ **do**

13: $\quad\quad$ $S_i \leftarrow \lambda \max_{\tilde{\mathbf{t}}_j \in \tilde{\mathcal{T}}} \text{PMI}(\tilde{\mathbf{v}}_i; \tilde{\mathbf{t}}_j) - (1 - \lambda) \max_{\tilde{\mathbf{v}}_j \in \tilde{\mathcal{V}}_{\text{S}}} \text{PMI}(\tilde{\mathbf{v}}_i; \tilde{\mathbf{v}}_j)$

14: $\quad$ **end for**

15: $\quad$ $\mathbf{v}^\star \leftarrow \arg\max_{\mathbf{v}_i} S_i, \text{and } \mathcal{V}_{\text{S}} \leftarrow \mathcal{V}_{\text{S}} \cup \{\mathbf{v}^\star\}$

16: **end while**

17: $\mathcal{V}_{\text{keep}} \leftarrow \mathcal{V}_{\text{S}}$

18: **return** $\mathcal{V}_{\text{keep}}$

---

**Scoring complexity.** The scoring function in Eqn. (22) consists of two terms. The crossmodal term holds a complexity of $\mathcal{O}(N_V N_T)$, *i.e.*, all text tokens attend the scoring process. Yet, the aggregation of intra-modal term is based on the increasing tokens in the selected set $\mathcal{V}_{\text{S}}$, instead of all visual tokens in $\mathcal{V}$. Therefore, the intra-modal complexity holds $\mathcal{O}(N_V N_{\text{keep}})$, instead of $\mathcal{O}(N_V N_V)$. The overall complexity considering two terms is $\mathcal{O}(N_V \cdot \max(N_T, N_{\text{keep}}))$, and becomes $\mathcal{O}(N_V N_T)$ when crossmodal priority ($\lambda=1$).

**Selection complexity.** The selection step aims to choose the top $N_{\text{keep}}$ elements from $N_V$ candidates. A naive greedy implementation selects one element per step by scanning all $N_V$ candidates to find the maximum gain, resulting in a complexity of $\mathcal{O}(N_V N_{\text{keep}})$. In contrast, we maintain a heap over all $N_V$ candidates for efficiency: building the heap by a cost of $\mathcal{O}(N_V)$, then performing $N_{\text{keep}}$ retrievals, each of which costs $\mathcal{O}(\log N_V)$. Our heap selection complexity yields an overall complexity of $\mathcal{O}(N_V + N_{\text{keep}} \log N_V)$. We expect $N_{\text{keep}} \log N_V < N_V$ in settings of interest, implying a selection complexity of $\mathcal{O}(N_V)$.

Algorithm 1 illustrates our full pipeline. In experiments, we empirically verify that this faster inference framework results in comparable or better performance to SOTA baselines.

## 5 Experiments

### 5.1 Experimental Setup

**Settings.** We conduct experiments on *open-ended QA* datasets, involving GQA (Hudson & Manning, 2019) and MMVet (Yu et al., 2023), and broader *closed-form* datasets, including SQA (Lu et al., 2022) (multiple-choice in image subset), $\text{VQA}_{\text{text}}$ (Singh et al., 2019) (reference answer provided), $\text{MME}_{\text{P}}$ (Fu et al., 2025) (Yes/No) and POPE (Li et al., 2025a) (Yes/No). More descriptions are in App. D. The query type influences the choice of MI-Pruner hyperparameters due to the varying textual information density. Based on experimental results in the ablation study (Sec. 5.5), we select the optimal parameters for open-ended and closed-form datasets, and keep them fixed in the comparative experiments. The inference and evaluation follow the default settings and metrics. Specifically, we report F1 score ($\mathbf{P_{F1}}$) and Yes-rate ($\mathbf{P_{Yes}}$) for POPE as fine-grained measures and report the mean score over bootstrapping for MMVet, since it only include 218 samples. The MLLMs include LLaVA1.5 (Liu et al., 2024a), Qwen2VL (Wang et al., 2024) and

latest Qwen3VL (Bai et al., 2025), all in greedy sampling and SDPA attention, more details in App. B.3. For generalization, we also apply our method to Video-LLaVA-7B (Lin et al., 2024) and test on TGIF-QA (Jang et al., 2017), MSVD-QA (Xu et al., 2017) and MSRVTT-QA (Xu et al., 2017) datasets. All experiments are conducted on a single NVIDIA A100 GPU with batch-size=1.

Table 1: **Performance comparison of different pruning methods on LLaVA1.5-7B.** We use color to distinguish best and second best. For POPE, a higher F1 score ($P_{F1}$) and a balanced Yes-ratio ($P_{Yes}$) around 0.5 indicate better performance.

| Methods | GQA | SQA | VQA$_{text}$ | MMVet | MME$_P$ | POPE | $P_{F1}$ | $P_{Yes}$ | Avg. | Δ% |
|---|---|---|---|---|---|---|---|---|---|---|
| *LLaVA1.5-7B* | 61.97 | 66.80 | 59.09 | 30.46 | 1509.13 | 85.18 | 0.86 | 0.56 | 60.70 | 0.00% |
| *keep 64 (88.9%↓)* | | | | | | | | | | |
| Random | 55.45 | 67.34 | 49.54 | 23.60 | 1274.27 | 79.82 | 0.76 | 0.34 | 55.15 | -9.14% |
| Similarity | 54.63 | 68.47 | 50.30 | 24.04 | 1286.66 | 84.24 | 0.83 | 0.42 | 56.34 | -7.19% |
| FastV (Chen et al., 2024a) ECCV'24 | 50.80 | 51.10 | 47.80 | 24.99 | 1019.60 | 48.50 | 0.48 | 0.35 | 44.64 | -26.46% |
| SparseVLM (Zhang et al., 2025f) ICML'25 | 52.70 | 62.20 | 51.80 | 23.04 | 1221.10 | 75.45 | 0.75 | 0.37 | 53.04 | -12.62% |
| VisionZip (Yang et al., 2025a) CVPR'25 | 55.10 | 69.00 | 55.50 | 31.04 | 1365.60 | 77.34 | 0.76 | 0.37 | 57.60 | -5.11% |
| VisPruner (Zhang et al., 2025c) ICCV'25 | 55.40 | 69.10 | 55.80 | 32.09 | 1369.90 | 82.61 | 0.80 | 0.38 | 59.00 | -2.80% |
| DART (Wen et al., 2025b) EMNLP'25 | 55.90 | 68.86 | 54.40 | 26.01 | 1273.30 | 73.90 | 0.73 | 0.33 | 55.81 | -8.05% |
| HoloV (Zou et al., 2025) NeurIPS'25 | 55.30 | 68.62 | 55.40 | 30.40 | 1367.11 | 80.30 | 0.80 | 0.37 | 58.00 | -4.44% |
| ApET (Ma et al., 2026) CVPR'26 | 56.85 | 68.90 | 53.00 | 27.63 | 1270.20 | 84.40 | 0.84 | 0.39 | 58.16 | -4.19% |
| Script (Yang et al., 2025d) TMLR'26 | 59.28 | 68.65 | 55.20 | 29.96 | 1412.08 | 86.95 | 0.85 | 0.41 | 60.01 | -1.14% |
| **MI-Attention** | 57.01 | 69.20 | 55.90 | 31.16 | 1428.03 | 85.04 | 0.84 | 0.41 | 59.66 | -1.71% |
| **MI-Pruner** | 56.88 | 69.81 | 54.91 | 28.12 | 1381.35 | 84.91 | 0.83 | 0.43 | 58.93 | -2.91% |
| *keep 32 (94.4%↓)* | | | | | | | | | | |
| Random | 52.58 | 67.33 | 47.20 | 21.72 | 1176.85 | 73.23 | 0.66 | 0.30 | 52.41 | -13.65% |
| Similarity | 51.46 | 67.48 | 47.78 | 20.05 | 1141.68 | 80.07 | 0.78 | 0.42 | 53.37 | -12.08% |
| FastV (Chen et al., 2024a) ECCV'24 | 41.50 | 42.60 | 42.50 | 20.55 | 884.60 | 33.50 | 0.33 | 0.34 | 36.13 | -40.48% |
| SparseVLM (Zhang et al., 2025f) ICML'25 | 48.30 | 57.30 | 46.10 | 18.81 | 1046.70 | 68.90 | 0.68 | 0.36 | 47.88 | -21.12% |
| VisionZip (Yang et al., 2025a) CVPR'25 | 51.80 | 68.80 | 53.10 | 25.43 | 1247.40 | 68.92 | 0.68 | 0.37 | 53.61 | -11.68% |
| VisPruner (Zhang et al., 2025c) ICCV'25 | 52.20 | 69.20 | 53.90 | 28.77 | 1271.00 | 77.83 | 0.73 | 0.32 | 56.38 | -7.12% |
| DART (Wen et al., 2025b) EMNLP'25 | 52.77 | 68.76 | 52.20 | 24.91 | 1273.30 | 69.13 | 0.69 | 0.31 | 53.55 | -11.77% |
| HoloV (Zou et al., 2025) NeurIPS'25 | 52.46 | 68.86 | 53.37 | 28.30 | 1268.55 | 81.20 | 0.81 | 0.38 | 56.84 | -6.36% |
| Script (Yang et al., 2025d) TMLR'26 | 57.58 | 68.75 | 53.10 | 27.57 | 1338.27 | 86.77 | - | - | 58.75 | -3.21% |
| **MI-Attention** | 54.19 | 69.11 | 54.07 | 28.91 | 1344.16 | 82.81 | 0.81 | 0.39 | 57.82 | -4.75% |
| **MI-Pruner** | 55.78 | 69.26 | 52.63 | 24.46 | 1307.72 | 83.18 | 0.81 | 0.41 | 57.06 | -5.99% |

**Comparison methods.** Our comparison involves the baseline with full tokens, salience- and subset-based methods and two naive approaches. For LLaVA1.5, we compare with *(i) salience-based methods* FastV (Chen et al., 2024a), SparseVLM (Zhang et al., 2025f), VisionZip (Yang et al., 2025a) and VisPruner (Zhang et al., 2025c), *(ii) subset-based methods* DART (Wen et al., 2025b), HoloV (Zou et al., 2025), ApeT (Ma et al., 2026) **and Script (Yang et al., 2025d)**, and *(iii) naive approaches*, where "Random" denotes averaging three independent runs with randomly selected tokens, and "Similarity" refers to cosine similarity between vision and text tokens (Eqn. (14)). As a variation, **MI-Attention** combines attention from the vision encoder (round-1) and MI-guided pruning (round-2). Since existing methods haven't released codes on Qwen3VL, we establish a general "Attention" baseline from VisionZip.

## 5.2 Main Results

**LLaVA1.5.** Following the well-established benchmark (Yang et al., 2025a; Zhang et al., 2025c), we report the pruning effects on LLaVA1.5-7B in Tab. 1. As observed, **MI-Pruner** and **MI-Attention** achieve SOTA performance in most benchmarks. We additionally report the performance of random pruning, which includes no importance or similarity measure. Surprisingly, the well-known methods FastV and SparseVLM don't really outperform "Random" as illustrated in Wang et al. (2025b); Wen et al. (2025b;a), underscoring the necessity of a solid theoretical ground. Moreover, most comparison methods significantly reduce the "Yes"-response proportion in POPE under 0.40, likely due to mistakenly pruning image tokens corresponding to the queried objects. In contrast, our method maintains a more balanced Yes/No distribution by preserving queried information in image tokens. In Fig. 1, we visualize the pruning effects of representative methods working in the projection space (ours), in LLMs (SparseVLM) and in vision encoders (VisPruner). **MI-**

**Pruner** consistently catches the queried region in three budgets. Notably, our approach does not rely on collecting attention maps. This non-intrusive characteristic makes it more suitable for practical deployment.

**Qwen2VL and Qwen3VL series.** Despite the model-agnostic nature, we further evaluate our method on advanced Qwen-series with dynamic resolutions. For Qwen2VL series, we show performance on different generations: 2VL and 2.5VL in Fig. 4. The visual cues in VisionZip depend on specific vision encoders, which limit its generalization capacity. In comparison, our method outperforms both subset-based DART and salience-based VisionZip and FastV. For the latest Qwen3VL series, we illustrate two scales 2B and 8B in Tab. 2, where "Attention" is im-

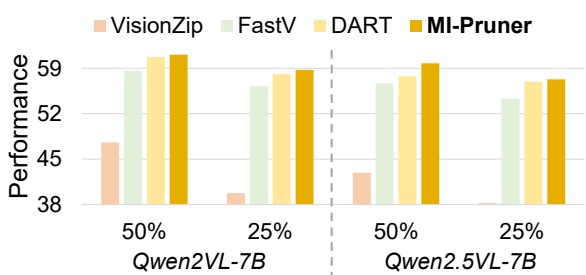

Figure 4: **Performance on Qwen2VL series (GQA).**

plemented as the attention sum-up from other image tokens, to avoid the specific design of [CLS] or hyper-parameters of existing methods (Yang et al., 2025a; Zhang et al., 2025c). Meanwhile, the visualization of pruning effects can be found in Fig. 5. By design, high-attention tokens produced by the vision encoder are biased toward generic salient features, such as faces or logos, which are pivotal for classification. However, such prompt-agnostic pruning exhibits reduced robustness when the query lies in non-salient or fine-grained regions that fall outside these pre-defined regions of interest. For instance, the Attention method keeps the facial landmark in Fig. 5a while pruning the queried cars. The scattered patches among the swimmer and the swimming pool in Fig. 5b increase the difficulty of multimodal reasoning. In comparison, **MI-Pruner** provides semantically complete patches aligned with the query.

**Efficiency analysis.** The big-O complexity is analyzed in Sec. 4.3. Here, we report the GPU memory and latency on POPE in Tab. 3. Since SparseVLM operates in LLMs, its efficiency benefits are suboptimal. Despite the early stage in vision encoder, VisPruner is still held back by its attention collection. As a diversity-based method, CDPruner compromises the efficiency due to MAP inference. Despite a slight latency increase when intra-model redundancy is considered ($\lambda \neq 1$), **MI-Pruner** delivers the best inference efficiency under both configurations.

### 5.3 Video Understanding Results

Following previous work (Yang et al., 2025a), we report video-QA pruning results on Video-LLaVA (Lin et al., 2024), which processes 8 frames with $256 \times 8 = 2048$ visual tokens. Tab. 4, compares ours with FastV and VisPruner under 227 (11.08% retained) and 114 (5.57% retained) video tokens budgets, scored by GPT-3.5-Turbo (OpenAI, 2023). **MI-Pruner** achieves average 93.19% accuracy across three datasets with merely 114 visual tokens, which indicates our robustness on video pruning.

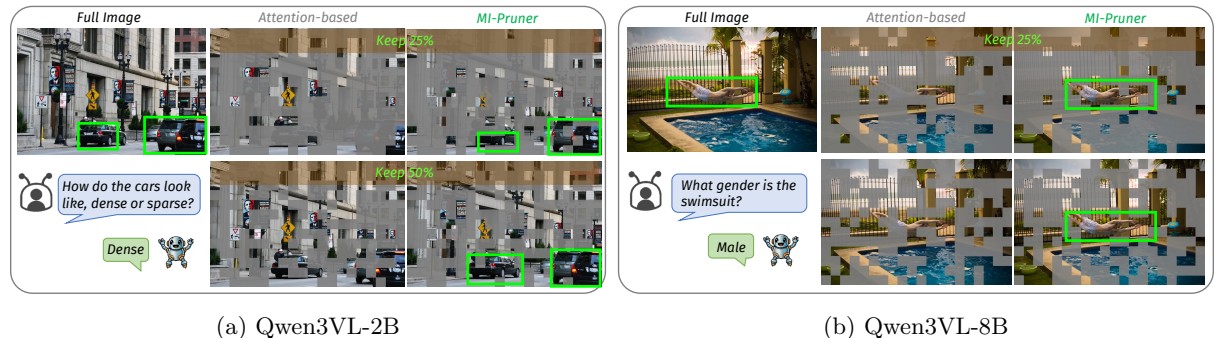

(a) Qwen3VL-2B    (b) Qwen3VL-8B

Figure 5: **Pruning visualization on Qwen3VL.** MI-Pruner retains semantic-relative patches regarding prompts adaptively, while salience-based methods tend to focus on logos and faces regardless of prompts.

Table 2: **Performance comparison of different pruning methods on Qwen3VL-2B and -8B.** "Attention" refers to the sum of attention scores from other image tokens. **MI-Pruner** and **MI-Attention** achieve SOTA performance for both scales.

| Methods | GQA | MME$_P$ | POPE | P$_{F1}$ | P$_{Yes}$ | Methods | GQA | MME$_P$ | POPE | P$_{F1}$ | P$_{Yes}$ |
|---|---|---|---|---|---|---|---|---|---|---|---|
| *Qwen3VL-2B* | 59.55 | 1494.98 | 89.89 | 0.90 | 0.47 | *Qwen3VL-8B* | 61.53 | 1730.47 | 88.84 | 0.88 | 0.44 |
| *keep 50%* | | | | | | | | | | | |
| Random | 49.96 | 1438.19 | 88.76 | 0.88 | 0.46 | Random | 53.02 | 1422.31 | 88.25 | 0.87 | 0.43 |
| Similarity | 50.87 | 1454.34 | 89.26 | 0.89 | 0.48 | Similarity | 53.17 | 1461.25 | 88.95 | 0.88 | 0.45 |
| Attention | 50.09 | 1462.74 | 89.54 | 0.89 | 0.47 | Attention | 52.93 | 1462.84 | 88.85 | 0.88 | 0.44 |
| **MI-Pruner** | 51.30 | 1478.26 | 89.78 | 0.90 | 0.48 | **MI-Pruner** | 53.48 | 1493.04 | 89.22 | 0.89 | 0.45 |
| **MI-Attention** | 51.11 | 1493.20 | 89.73 | 0.89 | 0.48 | **MI-Attention** | 53.23 | 1482.10 | 89.00 | 0.88 | 0.44 |
| *keep 25%* | | | | | | | | | | | |
| Random | 47.15 | 1398.52 | 85.59 | 0.85 | 0.42 | Random | 49.47 | 1256.18 | 85.64 | 0.84 | 0.41 |
| Similarity | 48.43 | 1372.50 | 88.28 | 0.88 | 0.47 | Similarity | 50.61 | 1338.28 | 87.98 | 0.88 | 0.45 |
| Attention | 46.59 | 1158.90 | 86.39 | 0.85 | 0.41 | Attention | 49.00 | 1293.91 | 87.74 | 0.87 | 0.43 |
| **MI-Pruner** | 49.63 | 1447.14 | 89.51 | 0.89 | 0.48 | **MI-Pruner** | 51.42 | 1369.58 | 88.82 | 0.88 | 0.45 |
| **MI-Attention** | 48.53 | 1459.36 | 89.38 | 0.89 | 0.46 | **MI-Attention** | 50.72 | 1415.71 | 88.82 | 0.88 | 0.44 |

Table 3: Efficiency comparison on LLaVA1.5-7B.

| Methods | Mem (GB) | Latency (ms) | Mem (GB) | Latency (ms) |
|---|---|---|---|---|
| | *keep 128* | | *keep 64* | |
| SparseVLM ICML'25 | 18.08 | 96.36$_{\pm0.35}$ | 18.11 | 93.67$_{\pm0.31}$ |
| VisPruner CVPR'25 | 14.35 | 92.04$_{\pm0.44}$ | 14.35 | 89.48$_{\pm0.32}$ |
| CDPruner NeurIPS'25 | 14.73 | 111.12$_{\pm0.65}$ | 14.65 | 91.41$_{\pm0.41}$ |
| DART EMNLP'25 | 14.05 | 92.62$_{\pm0.39}$ | 13.96 | 88.41$_{\pm0.37}$ |
| **MI-Pruner**$_{\lambda=0.5}$ | 14.01 | 98.92$_{\pm0.40}$ | 13.93 | 84.95$_{\pm0.34}$ |
| **MI-Pruner**$_{\lambda=1}$ | 14.01 | 82.08$_{\pm0.32}$ | 13.93 | 77.06$_{\pm0.36}$ |
| Script TMLR'25 | 14.73 | 126.11$_{\pm0.47}$ | 14.65 | 100.04$_{\pm0.38}$ |

Table 4: Performance on Video-LLaVA-7B.

| Methods | TGIF | | MSVD | | MSRVTT | |
|---|---|---|---|---|---|---|
| | *Acc* | *Score* | *Acc* | *Score* | *Acc* | *Score* |
| *Video-LLaVA* | 18.90 | 2.54 | 72.00 | 3.95 | 57.50 | 3.50 |
| *keep 227* | | | | | | |
| FastV ECCV'24 | 14.30 | 2.42 | 68.90 | 3.90 | 53.00 | 3.40 |
| VisPruner ICCV'25 | 15.90 | 2.41 | 69.30 | 3.92 | 55.60 | 3.45 |
| **MI-Pruner** | 15.70 | 2.44 | 70.60 | 3.94 | 56.40 | 3.46 |
| *keep 114* | | | | | | |
| FastV ECCV'24 | 10.60 | 2.29 | 64.10 | 3.78 | 52.40 | 3.39 |
| VisPruner ICCV'25 | 14.10 | 2.35 | 65.40 | 3.79 | 54.10 | 3.41 |
| **MI-Pruner** | 13.10 | 2.40 | 69.90 | 3.92 | 55.30 | 3.49 |

## 5.4 Discussions

**Relationship with attention mechanism.** The inner product and softmax operations in **MI-Pruner** also appear in the attention mechanism (Vaswani et al., 2017). However, our algorithm is distinct from the attention module from 3 perspectives. *(i) Implementation:* Our scoring function is conducted just once in the projection space, after which the attention burden in LLM blocks is largely reduced. *(ii) Training-free nature:* We require no training step to learn the QKV matrices as the attention module; therefore, **MI-Pruner** is versatile and ready to be applicable to any pretrained MLLMs. *(iii) Theoretical ground:* Our importance measure is derived from Mutual Information (Jaynes, 1957), which is a traditional metric in information theory (details in Section 3.3) to quantify the statistical dependence among tokens. Framing token selection issue in a submodular framework, **MI-Pruner** achieves SOTA performance from information-theoretic motivation , not from a scaling law (Vaswani et al., 2017).

**Contributions and benefits.** Once conducted in the projection space, we decrease the visual attention cost in each LLM block from $N_V^2$ to $N_{keep}^2$, where $N_{keep} \ll N_V$. For instance, LLaVA1.5-7B (Liu et al., 2024a) employs 32 LLM blocks and 576 image tokens. The visual attention computation scales with $32 \cdot 576^2$ by default, but can be decreased to $32 \cdot 32^2$ (9 million↓) in our 32 setting. Experiments validate our performance (Tab. 1 and 2) and efficiency (Tab. 3). Moreover, **MI-Pruner** is easy to implement and supports various attention cores, like FlashAttention (see Tab. 7 in appendix). Rooted in submodularity, our MI-based token selection is theoretically grounded and offers enhanced interpretability. Unlike previous visual-only token reduction methods (Zhang et al., 2025c; Yang et al., 2025a), MI-Pruner explicitly accounts for both crossmodal and intra-modal dependencies, thereby demonstrating superior robustness. The visualization presented in Fig. 1 and Fig. 5 validates the interpretability of our approach over existing baselines. Among prior MI-based methods, TrimTokenator (Zhang et al., 2025a) adopts L2-norm proxy for a two-stage filtering, while AutoPrune (Wang et al., 2025a) collects attention maps and implicitly leverages MI for layer-wise token budgets. Another work MMTok (Dong et al.,

2026) relies on cosine-based coverage, not explicitly falling into MI frameworks. In comparison, MI-Pruner directly estimates PMI in an attention-free manner, and conducts greedy search once before LLMs. Empirical comparison can be found in Tab. 8, where we adopt the results of TrimTokenator (Zhang et al., 2025a) from the original paper.

### 5.5 Ablation Study

**Aggregation strategies.** We adopt max aggregation (Eqn. (19)-(20)) in the MI-based scoring function. Fig. 6a compares the performance of max and average aggregation under different token retention settings (128, 64 and 32 tokens). Empirically, the max aggregation consistently outperforms average aggregation, likely because it better preserves the discriminative characteristics of the queried object.

**Hyperparameters.** The temperature $\tau$ in Eqn. (14) controls the sharpness of similarity scores. Fig. 6b evaluates three groups of $\tau$ values from 0.01 to 0.5, showing that $\tau$=0.1 achieves the best performance. In addition, the factor $\lambda$ in Eqn. (22) modulates the scoring function to be crossmodal-only, intra-modal-only or a combination of both. As a task-related parameter, the cross- and intra-modal combination ($\lambda$=0.5) suits open-ended datasets (like GQA), and the crossmodal guided pruning is sufficient for closed-form datasets (like SQA) in Fig. 6c, since the prompt explicitly encodes the ground-truth cue in choices. The highest metrics are annotated for distinguishing.

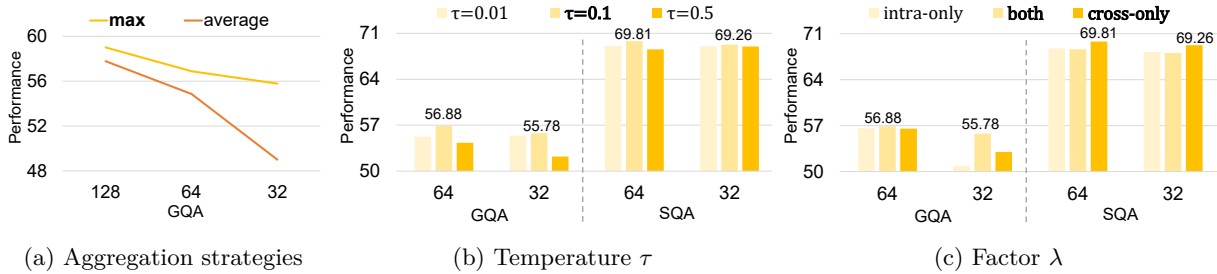

(a) Aggregation strategies      (b) Temperature $\tau$      (c) Factor $\lambda$

Figure 6: **Ablation study on LLaVA1.5-7B** (x-axis: the number of tokens, y-axis: performance scores).

## 6 Conclusion

We propose **MI-Pruner**, a plug-in and training-free visual pruning method. Formulating token pruning as a subset selection problem, we propose a relevance score based on Mutual Information. To avoid the reliance on attention maps, **MI-Pruner** works surgically in the projection space, providing a principled and efficient framework for token reduction. Experiments and visualization verified our robustness and efficiency. The surgical manner enables our method to be applied to any off-the-shelf MLLMs with the most favorable inference speedup.

**Limitations.** We assume equal visual probability and conditional independence. Future work involving an appropriate visual prior is expected to refine our results. Moreover, our pruning is conducted only for visual tokens, yet prompts also contain low-information tokens, *e.g.*, `"_a"` and `"_please"`. The textual token pruning plays a significant role in "needle in a haystack" and "conflict detection" tasks.

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

# Appendix

## A    Theory

### A.1    Definition

**Definition 4** (Shannon Entropy (Kraskov et al., 2004; Gray, 2011; Gallager, 1968))**.** *The Shannon entropy $H(X)$ of a random variable $X$ measures the average uncertainty or information content associated with $X$:*

$$H(X) = -\sum_{x \in \mathcal{X}} p(x) \log p(x). \tag{25}$$

**Definition 5** (Conditional Entropy (Kraskov et al., 2004; Gray, 2011; Gallager, 1968))**.** *Given two random variables $X$ and $Y$, the conditional entropy $H(X|Y)$ quantifies the amount of information needed to describe $X$ given that known $Y$:*

$$H(X|Y) = -\sum_{u \in \mathcal{X}, y \in \mathcal{Y}} p(x, y) \log p(x|y). \tag{26}$$

**Definition 6** (Conditional Mutual Information (Gallager, 1968))**.** *The Conditional Mutual Information $\mathrm{MI}(X; Y \mid Z)$ measures the reduction in uncertainty of a random variable $X$ given $Z$ when additionally knowing $Y$:*

$$\mathrm{MI}(X; Y \mid Z) = H(X \mid Z) - H(X \mid Y, Z) \tag{27}$$

$$= \sum_{x \in \mathcal{X}, \, y \in \mathcal{Y}, \, z \in \mathcal{Z}} p(x, y, z) \log \frac{p(x \mid y, z)}{p(x \mid z)}. \tag{28}$$

**Definition 7** (Submodular Function (Fujishige, 2005)). *Let $X$ be a finite set. A set function $f : 2^X \to \mathbb{R}$ is called submodular if for every $A, B \subseteq X$, we have:*

$$f(A \cup B) + f(A \cap B) \leq f(A) + f(B). \tag{29}$$

An equivalent and often more intuitive definition is based on the *marginal gain*, as shown in the main paper.

## A.2 Conditional Independence

> **Assumption:** The candidate $\mathbf{v}_i$ is independent of the selected set $\mathcal{V}_S$ conditioned on $\mathcal{T}$.
> *This assumption helps to maintain submodularity and derive tractable marginal gains.*

Mutual Information is widely recognized for exhibiting submodularity (the property of diminishing returns) (Iyer et al., 2021; Krause et al., 2008), providing a principled foundation for efficient subset selection.

To rigorously adapt MI-based submodular optimization to the token selection task, we bring the Naive Bayes assumption as the conditional independence between $\mathbf{v}_i$ and $\mathcal{V}_S$ based on $\mathcal{T}$. From the view of probability and Mutual Information, it holds:

$$p(\mathbf{v}_i, \mathcal{V}_S \mid \mathcal{T}) = p(\mathbf{v}_i \mid \mathcal{T})p(\mathcal{V}_S \mid \mathcal{T}), \tag{30}$$

$$\mathrm{MI}(\mathbf{v}_i; \mathcal{V}_S \mid \mathcal{T}) = 0. \tag{31}$$

This assumption has two immediate consequences: (i) *submodular* guarantees for the property of diminishing returns; (ii) a tractable derivation of the *marginal gain*.

**Submodularity.** The conditional independence assumes that a single patch is sufficiently representative of a semantic unit. A counterexample is that, if a semantic concept strictly requires the joint presence of multiple patches (e.g., both a left and a right part), then adding the remaining patch at a later stage could violate the diminishing-returns property. Nevertheless, the assumption of semantic sufficiency is well justified in practice. For example, a single patch depicting a "wheel" or a "grille" is often sufficient to convey the semantic concept of "`_car`". Based on this assumption, we can robustly treat the visual token selection process as a submodular maximization problem, ensuring both theoretical consistency and computational efficiency.

**Marginal gains.** Since accurately estimating the joint distribution involving $\mathcal{V}_S$ is high-dimensional infeasible, we approximate the conditional Mutual Information by the marginal Mutual Information. According to Def. 1, Mutual Information can be expressed in terms of entropy:

$$\mathrm{MI}(X; Y) = H(Y) - H(Y \mid X). \tag{32}$$

Here, we give the proof from Eqn. (11) to Eqn. (12):

$$\mathrm{MI}(\mathcal{V}_S \cup \{\mathbf{v}_i\}; \mathcal{T}) - \mathrm{MI}(\mathcal{V}_S; \mathcal{T}) \tag{33}$$

$$= \left[ \cancel{H(\mathcal{T})} - H(\mathcal{T} \mid \mathcal{V}_S \cup \{\mathbf{v}_i\}) \right] - \left[ \cancel{H(\mathcal{T})} - H(\mathcal{T} \mid \mathcal{V}_S) \right] \tag{34}$$

$$= H(\mathcal{T} \mid \mathcal{V}_S) - H(\mathcal{T} \mid \mathcal{V}_S \cup \{\mathbf{v}_i\}) \tag{35}$$

$$= \mathrm{MI}(\mathbf{v}_i; \mathcal{T} \mid \mathcal{V}_S). \tag{36}$$

Further, this conditional MI can be written as:

$$\mathrm{MI}(\mathbf{v}_i; \mathcal{T} \mid \mathcal{V}_S) = \underbrace{\mathrm{MI}(\mathbf{v}_i; \mathcal{T}, \mathcal{V}_S)}_{\mathrm{MI}^{\mathrm{cross}}} - \underbrace{\mathrm{MI}(\mathbf{v}_i; \mathcal{V}_S)}_{\mathrm{MI}^{\mathrm{self}}} \tag{37}$$

$$= \mathrm{MI}(\mathbf{v}_i; \mathcal{V}_S \mid \mathcal{T}) + \mathrm{MI}(\mathbf{v}_i; \mathcal{T}) - \mathrm{MI}(\mathbf{v}_i; \mathcal{V}_S). \tag{38}$$

However, the crossmodal term is intractable, which causes trouble in deriving the marginal gain $\Delta f$. Again, we leverage the conditional independence assumption, and substitute Eqn. (31) into Eqn. (38). Since

$\underline{\text{MI}(\mathbf{v}_i; \mathcal{V}_\text{S} \mid \mathcal{T})} = 0$, we obtain an estimator:

$$\widehat{\text{MI}}(\mathbf{v}_i; \mathcal{T} \mid \mathcal{V}_\text{S}) = \text{MI}(\mathbf{v}_i; \mathcal{T}) - \text{MI}(\mathbf{v}_i; \mathcal{V}_\text{S}). \tag{39}$$

The assumption $\text{MI}(\mathbf{v}_i; \mathcal{V}_\text{S} \mid \mathcal{T}) = 0$ implies that the crossmodal relevance is much stronger than the internal one, *i.e.*, $\text{MI}^\text{cross} \to \text{MI}(\mathbf{v}_i, \mathcal{T})$. It is empirically justified under high pruning rates (less than 50% retained), where sparse selection leads to trivial intra-modal overlap.

For practical implementation, we allow balancing the crossmodal term and the intra-model term with a factor $\lambda$. As shown in Eqn. (21), we derive two scoring functions respectively and combine them with $\lambda$.

## A.3 Maximal Aggregation

Our max aggregation over $\mathcal{T}$ and $\mathcal{V}_\text{select}$ is motivated by the LogSumExp (LSE) operator, defined as

$$\text{LSE}(z_1, \ldots, z_N) := \log \left( \sum_{j=1}^{N} \exp(z_j) \right). \tag{40}$$

As a smooth upper approximation to the maximum, the LSE operator satisfies the following bounds:

$$\max_j z_j \ \leq \ \text{LSE}(z_1, \ldots, z_N) \ \leq \ \max_j z_j + \log N. \tag{41}$$

Here, we justify the maximal aggregation as a robust proxy for the overall crossmodal relevance. For simplification, we omit $\tilde{\ }$ of $\mathbf{v}_i, \mathbf{t}_j$ Theoretically, let $\mathbf{z}_j = \text{PMI}(\mathbf{v}_i; \mathbf{t}_j)$ be the log-likelihood ratio for each token pair:

$$\mathbf{z}_j = \log \frac{p(\mathbf{t}_j \mid \mathbf{v}_i)}{p(\mathbf{t}_j)}. \tag{42}$$

The relationship between the peak signal and the total likelihood ratio can be characterized by the LSE operator:

$$\text{LSE}(\mathbf{z}_j) = \log \left( \sum_{j=1}^{N_T} \exp(\log \frac{p(\mathbf{t}_j \mid \mathbf{v}_i)}{p(\mathbf{t}_j)}) \right) \tag{43}$$

$$= \log \left( \sum_{j=1}^{N_T} \frac{p(\mathbf{t}_j \mid \mathbf{v}_i)}{p(\mathbf{t}_j)} \right). \tag{44}$$

Substitute Eqn. (44) into Eqn. (41), we obtain the lower bound of LSE, and exponentiate both sides:

$$\max_j (\log \frac{p(\mathbf{t}_j \mid \mathbf{v}_i)}{p(\mathbf{t}_j)}) \leq \log \left( \sum_{j=1}^{N_T} \frac{p(\mathbf{t}_j \mid \mathbf{v}_i)}{p(\mathbf{t}_j)} \right), \tag{45}$$

$$\max_j (\frac{p(\mathbf{t}_j \mid \mathbf{v}_i)}{p(\mathbf{t}_j)}) \leq \sum_{j=1}^{N_T} \frac{p(\mathbf{t}_j \mid \mathbf{v}_i)}{p(\mathbf{t}_j)}. \tag{46}$$

It's also expressed as:

$$\max_j (\text{PMI}(\mathbf{v}_i; \mathbf{t}_j)) \leq \sum_{j=1}^{N_T} \text{PMI}(\mathbf{v}_i; \mathbf{t}_j). \tag{47}$$

Maximal aggregation serves as a tight lower bound for the sum of PMI, which has the same optimization direction as the global aggregation. In practice, crossmodal correspondence exhibits inherent sparsity: within a sentence, typically only a few core tokens are semantically grounded in a specific visual region. Therefore, averaging these sparse, high-intensity signals with numerous irrelevant tokens leads to a noisy dilution of the relevance signal. In contrast, the max aggregation, analogous to an $L_\infty$-type norm of the association scores, captures the most salient semantic tokens and preserves the alignment measure from noise.

### A.4 Marginal Distribution

**Assumption:** Each image patch occurs with equal probability $p(\tilde{\mathbf{v}}_i) = \frac{1}{N_V}$.
*This assumption helps to derive the marginal text probability $p(\tilde{\mathbf{t}}_j)$.*

In this section, we recap the Law of Total Probability (Jaynes, 1957) and then derive $p(\tilde{\mathbf{t}}_j)$ in Eqn. (16).

**Proposition A.1** (Law of Total Probability (Jaynes, 1957)). *Let $A$ be an event and let $\{B_i\}_{i=1}^{N}$ be a set of mutually exclusive and exhaustive events. The Law of Total Probability indicates:*

$$p(A) = \sum_{i=1}^{N} p(A \mid B_i)\, p(B_i). \tag{48}$$

Since the mapping between a patch and a token is deterministic, $\{\tilde{\mathbf{v}}_i\}_{i=0}^{N_V}$ are exclusive and exhaustive. We derive the marginal probability by substituting $A, B_i$ with $\tilde{\mathbf{t}}_j, \tilde{\mathbf{v}}_i$:

$$p(\tilde{\mathbf{t}}_j) = \sum_{i=1}^{N_V} p(\tilde{\mathbf{t}}_j \mid \tilde{\mathbf{v}}_i)\, p(\tilde{\mathbf{v}}_i). \tag{49}$$

Assuming that $N_V$ patches are equally probable, *i.e.*, the prior probability $p(\tilde{\mathbf{v}}_i) = \frac{1}{N_V}$, Eqn. (16) is formulated as:

$$p(\tilde{\mathbf{t}}_j) = \sum_{i=1}^{N_V} p(\tilde{\mathbf{t}}_j \mid \tilde{\mathbf{v}}_i) \underbrace{\frac{1}{N_V}}_{p(\tilde{\mathbf{v}}_i)} . \tag{50}$$

According to the Principle of Maximum Entropy (Gray, 2011), our uniformity is the most unbiased assumption, ensuring that the selection process is driven solely by the observed Mutual Information rather than predefined spatial heuristics. It is non-trivial to obtain the appropriate prior for vision tokens. We admit potential solutions of constructing an estimator from the validation set, or linking with the patch position based on photography composition techniques (*e.g.*, Rule of Thirds and Golden Ratio). The estimation of the prior probability is set as our future work.

## B More Experiments

### B.1 POPE Results

Tab. 1 and 2 show the average results of POPE. In the appendix, we report detailed performance under random/popular/adversarial settings, see Tab. 11 and 12. The latest Qwen3VL outperforms LLaVA1.5 across three settings, while both fall short of adversarial settings. The results after pruning hold the same tendency. Nevertheless, our method demonstrates remarkable resilience in adversarial settings, outperforming the baselines by a significant margin under different budgets. This success highlights our approach's acute sensitivity in capturing essential visual features regarding the prompt.

### B.2 MinMax Normalizarion

Based on Boltzmann distributions, we compute similarity matrices and apply softmax in Eqn. (15). In the appendix, we conduct an ablation study to investigate the impact of linear normalization. Specifically, we implement a two-stage MinMax normalization as follows:

$$\hat{\boldsymbol{\rho}}_{ij}^{\mathrm{s}} = \mathrm{MinMax}_j(\boldsymbol{\rho}_{ij}^{\mathrm{s}}) = \frac{\boldsymbol{\rho}_{ij}^{\mathrm{s}} - \mathrm{Min}_j(\boldsymbol{\rho}_{ij}^{\mathrm{s}})}{\mathrm{Max}_j(\boldsymbol{\rho}_{ij}^{\mathrm{s}}) - \mathrm{Min}_j(\boldsymbol{\rho}_{ij}^{\mathrm{s}})}, \tag{51}$$

$$p(\tilde{\mathbf{x}}_j \mid \tilde{\mathbf{v}}_i) = \mathrm{Normalize}_j(\hat{\boldsymbol{\rho}}_{ij}^{\mathrm{s}}) = \frac{\hat{\boldsymbol{\rho}}_{ij}^{\mathrm{s}}}{\sum_{k=1}^{N} \hat{\boldsymbol{\rho}}_{ij}^{\mathrm{s}}}. \tag{52}$$

Here, $\hat{\rho}_{ij}^{\mathrm{s}}$ first maps the raw similarity scores to the $[0,1]$ range, then Eqn. (52) ensures a valid probability distribution across the dimension $j$. As illustrated in Tab. 5, our softmax normalization outperforms MinMax and maintains its overall robust performance. We attribute this performance gap to the non-linear saliency of softmax and its robustness to outliers.

Table 5: Ablation study on Normalization (LLaVA1.5-7B).

|  | $\mathbf{GQA}_{64}$ | $\mathbf{GQA}_{32}$ | $\mathbf{SQA}_{64}$ | $\mathbf{SQA}_{32}$ |
|---|---|---|---|---|
| MinMax | 55.37 | 52.95 | 67.56 | 67.01 |
| Softmax | 56.88 | 55.78 | 69.81 | 69.26 |

## B.3 Large-scale Models and Diverse Settings

To further demonstrate the robustness of our method, we report results on larger-scale models, including LLaVA1.5-13B and Qwen3VL-30B (Mixture-of-Experts). In Tab. 6, our method consistently outperforms random sampling and similarity-based pruning. Yet, we acknowledge that for the latest models (Bai et al., 2025), model-specific tailoring is indispensable. Qwen3VL employs deep-stack integration and adaptive padding mechanisms, which demand dedicated designs to maintain structural integrity and performance during pruning. By providing a principled foundation, our method facilitates future research and remains highly adaptable to next-generation multimodal architectures.

Table 6: Results on large-scale models, LLaVA1.5-13B and Qwen3VL-30B.

| Methods | GQA | SQA | $\mathbf{VQA}_{\text{text}}$ | $\mathbf{MME}_{\text{P}}$ |
|---|---|---|---|---|
| *LLaVA1.5-13B* | 61.97 | 72.73 | 61.24 | 1524.19 |
| *keep 64* | | | | |
| Random | 53.45 | 70.85 | 50.74 | 1257.35 |
| cosine | 53.63 | 71.64 | 51.58 | 1338.78 |
| FastV | 53.70 | 56.80 | 47.10 | 1275.41 |
| SparseVLM | 50.60 | 69.00 | 22.70 | 1289.92 |
| **MI-Pruner** | 55.64 | 71.79 | 53.19 | 1378.49 |
| *Qwen3VL-30B* | 62.51 | 95.29 | 83.20 | 1815.29 |
| *keep 50%* | | | | |
| Random | 55.22 | 91.18 | 34.89 | 1755.03 |
| cosine | 56.47 | 88.65 | 29.96 | 1791.04 |
| Attention | 56.02 | 88.98 | 30.02 | 1799.54 |
| **MI-Pruner** | 56.66 | 92.27 | 37.65 | 1822.19 |

Table 7: Efficiency comparison on LLaVA1.5-7B.

| Methods | Mem (GB) | Latency (ms) |
|---|---|---|
| *keep 32* | | |
| SparseVLM | 18.12 | $93.33_{\pm 0.44}$ |
| VisPruner | 14.35 | $89.98_{\pm 0.39}$ |
| DART | 13.94 | $87.56_{\pm 0.47}$ |
| CDPruner | 14.61 | $83.22_{\pm 0.38}$ |
| **MI-Attention** | 14.35 | $77.36_{\pm 0.40}$ |
| **MI-Pruner**$_{\lambda=1}$ | 13.90 | $77.07_{\pm 0.32}$ |
| **MI-Pruner**$_{\lambda=1,\text{Flash}}$ | 13.90 | $75.97_{\pm 0.28}$ |
| **MI-Pruner**$_{\lambda=0.5}$ | 13.90 | $78.71_{\pm 0.34}$ |
| **MI-Pruner**$_{\lambda=0.5,\text{Flash}}$ | 13.90 | $78.00_{\pm 0.31}$ |

## B.4 Efficiency Analysis

We evaluate the end-to-end efficiency by tracking GPU memory usage and latency throughout the process, including visual encoding, LLM prefilling, and subsequent decoding. To ensure stable results, we conduct 10 warm-up runs and report the average GPU memory usage and latency over 30 repetitions. We extend Tab. 3 in the main paper to $N_{\text{keep}} = 32$ in Tab. 7, and report the results on FlashAttention2 (Dao, 2024). Among diversity-based methods, DART (Wen et al., 2025b) and CDPruner (Zhang et al., 2025d) are faster than salience-based methods, but less efficient than **MI-Pruner**. As analyzed in the original paper (Zhang et al., 2025d), the computational bottleneck of CDPruner lies in its DPP MAP inference, which incurs a complexity of $\mathcal{O}(N_V N_{\text{keep}}^2)$. In comparison, **MI-Pruner**'s complexity is $\mathcal{O}(N_V \cdot \max(N_T, N_{\text{keep}}))$ for full scoring and $\mathcal{O}(N_V N_T)$ for relevance-based sorting. Given that the instruction length $N_T$ is typically much smaller than the budget $N_{\text{keep}}$ in practice, our method offers superior computational efficiency. Applying our MI-based scores in VisPruner, **MI-Attention** achieves the same GPU memory usage as VisPruner while exhibiting lower latency. Finally, **MI-Pruner** speeds up the inference further with FlashAttention cores, showing good compatibility.

### B.5 More Results

Tab. 8 shows a comparison with other MI-based methods, where our approach exhibits comparable performance and the best efficiency. As acknowledged in the limitation, we adopt the uniform spatial prior assumption for generalization. We claim that such an unbiased prior shows most advantages over many complex alternatives. To provide an ablation study, we introduce a "central object" prior by a weight factor $\mathbf{w}_{ij}$ for LLaVA1.5-7B, where an image is partitioned into $24 \times 24$ tokens. Denote $\mathbf{v}_{ij}$ as an image token in the 2D feature map and $\mathbf{v}_{\text{center}}$ as the center, we have:

$$\mathbf{w}_{ij} = \exp\left(-\frac{\|\mathbf{v}_{ij} - \mathbf{v}_{\text{center}}\|^2}{2\sigma^2}\right), \tag{53}$$

$$p(\mathbf{v}_{ij}) = \text{softmax}(\mathbf{w}_{ij}), \tag{54}$$

where $\sigma$ controls the decay rate and is measured in tokens. Under the default uniform prior, $p(\mathbf{v}_{ij}) = \frac{1}{N_V}$. As shown in Tab. 9, the central prior doesn't improve the overall performance, especially for OCR-related tasks (*e.g.* TextVQA). The rationale behind it is that forcing an "object-centric" spatial prior makes MLLMs over-focus on the center and neglect the fine-grained details scattered across the edges. In comparison, the uniform setting introduces less bias to maintain the best performance. A more complicated spatial prior and a data-dependent tuning might help in some cases.

Moreover, Tab. 10 offers an ablation study on plain cosine *vs* PMI and MinMax normalization *vs* softmax. Specifically, the "cosine+TopK" setting directly selects visual tokens with the highest crossmodal cosine similarity, which is called "Similarity" in Tab. 1 and Tab. 2. A new setting "cosine+greedy" replaces the PMI estimator with the plain cosine in Eqn. (22) for greedy search. As observed, ours achieves the best performance on both models by adopting the PMI estimator and softmax normalization.

Table 8: **Extended comparison on MI-based methods (LLaVA1.5-7B, keep 64).**

|  | GQA | VQA$_{v2}$ | Vizwiz | MMMU | MMB | Latency (ms) |
|---|---|---|---|---|---|---|
| TrimTokenator (Zhang et al., 2025a) | - | - | - | 36.1 | 61.3 | - |
| AutoPrune (Wang et al., 2025a) | 57.7 | 74.8 | 53.6 | 36.0 | 63.6 | 89.6 |
| MMTok (Dong et al., 2026) | 58.3 | 75.2 | 53.1 | 36.1 | 61.2 | 77.7 |
| **MI-Pruner** | 56.9 | 75.0 | 53.1 | 36.3 | 61.9 | 77.1 |
| **MI-Attention** | 57.0 | 75.5 | 53.4 | 36.1 | 62.1 | 77.5 |

Table 9: **Extended ablation on spatial prior (LLaVA1.5-7B, keep 64).**

|  | GQA | SQA | VQA$_{v2}$ | TextVQA |
|---|---|---|---|---|
| Central$_{\sigma=2}$ | 55.8 | 68.0 | 74.8 | 52.1 |
| Central$_{\sigma=5}$ | 55.3 | 67.5 | 74.2 | 51.3 |
| **MI-Pruner** | 56.9 | 69.8 | 75.2 | 54.9 |

Table 10: **Extended ablation on PMI estimator and softmax normalization.**

| Methods | GQA | SQA | TextVQA | POPE | Methods | GQA | SQA | TextVQA | POPE |
|---|---|---|---|---|---|---|---|---|---|
| *LLaVA1.5-7B/keep 64* | | | | | *Qwen3VL-8B/keep 25%* | | | | |
| MinMax | 54.7 | 67.9 | 53.2 | 83.3 | MinMax | 51.5 | 83.2 | 72.9 | 88.0 |
| cosine+TopK | 54.6 | 68.5 | 50.3 | 84.2 | cosine+TopK | 50.6 | 83.9 | 73.5 | 88.0 |
| cosine+greedy | 56.0 | 67.8 | 54.2 | 83.9 | cosine+greedy | 50.7 | 83.5 | 72.7 | 87.9 |
| **MI-Pruner** | 56.8 | 69.8 | 54.9 | 84.9 | **MI-Pruner** | 53.5 | 84.0 | 73.5 | 88.8 |

### B.6 Generalization

Our method is applicable to any Enc-MLP-Dec architectures, while previous methods rely on specific vision encoders. For instance, VisPruner (Zhang et al., 2025c) necessitates [CLS] as attention measures, therefore,

fails to adapt to the latest QwenVL series. Although VisionZip (Yang et al., 2025a) provides implementation for Qwen2.5VL, it suffers from a significant decline in instruction-following capacity at high pruning rates, as shown in Fig. 7. Due to its limited generalization and lack of theoretical guarantees, the model becomes increasingly prone to hallucinations under aggressive compression, *e.g.*, keeping 25% vision tokens.

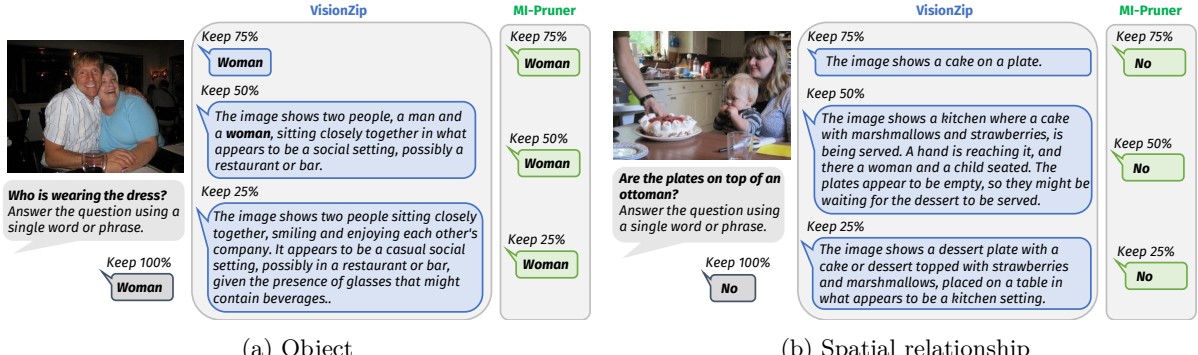

(a) Object  (b) Spatial relationship

Figure 7: Dialogue examples on Qwen2.5VL-7B (GQA).

Table 11: **POPE performance comparison under 3 settings on LLaVA1.5-7B.** We report accuracy (Acc), F1 scores and Yes-rate.

| Methods | *Random* | | | *Popular* | | | *Adversarial* | | | ***Average*** | | |
|---|---|---|---|---|---|---|---|---|---|---|---|---|
| | Acc ↑ | F1 ↑ | Yes | Acc ↑ | F1 ↑ | Yes | Acc ↑ | F1 ↑ | Yes | Acc ↑ | F1 ↑ | Yes |
| *LLaVA1.5-7B* | 89.60 | 0.90 | 0.51 | 86.20 | 0.87 | 0.55 | 79.73 | 0.82 | 0.61 | 85.18 | 0.86 | 0.56 |
| *keep 128* | | | | | | | | | | | | |
| Random | 85.23 | 0.83 | 0.37 | 84.07 | 0.82 | 0.38 | 80.80 | 0.79 | 0.41 | 83.37 | 0.81 | 0.39 |
| Similarity | 87.43 | 0.86 | 0.42 | 86.30 | 0.85 | 0.43 | 82.60 | 0.82 | 0.46 | 85.44 | 0.84 | 0.44 |
| **MI-Attention** | 88.31 | 0.87 | 0.41 | 86.63 | 0.86 | 0.42 | 84.11 | 0.83 | 0.44 | 86.35 | 0.85 | 0.42 |
| **MI-Pruner** | 88.43 | 0.87 | 0.42 | 86.90 | 0.86 | 0.43 | 84.07 | 0.83 | 0.46 | 86.47 | 0.85 | 0.44 |
| *keep 64* | | | | | | | | | | | | |
| Random | 80.53 | 0.76 | 0.33 | 80.57 | 0.77 | 0.33 | 78.37 | 0.75 | 0.36 | 79.82 | 0.76 | 0.34 |
| Similarity | 85.80 | 0.84 | 0.40 | 85.03 | 0.84 | 0.41 | 81.90 | 0.81 | 0.44 | 84.24 | 0.83 | 0.42 |
| **MI-Attention** | 86.53 | 0.85 | 0.40 | 85.60 | 0.84 | 0.40 | 83.00 | 0.82 | 0.43 | 85.04 | 0.84 | 0.41 |
| **MI-Pruner** | 86.83 | 0.85 | 0.41 | 85.63 | 0.84 | 0.42 | 82.27 | 0.81 | 0.45 | 84.91 | 0.83 | 0.43 |
| *keep 32* | | | | | | | | | | | | |
| Random | 74.93 | 0.67 | 0.26 | 75.43 | 0.68 | 0.28 | 73.23 | 0.66 | 0.30 | 73.23 | 0.66 | 0.30 |
| Similarity | 83.87 | 0.82 | 0.38 | 83.20 | 0.81 | 0.39 | 80.07 | 0.78 | 0.42 | 80.07 | 0.78 | 0.42 |
| **MI-Attention** | 84.37 | 0.82 | 0.37 | 83.23 | 0.81 | 0.39 | 80.83 | 0.79 | 0.41 | 82.81 | 0.81 | 0.39 |
| **MI-Pruner** | 85.07 | 0.83 | 0.39 | 83.87 | 0.82 | 0.40 | 80.60 | 0.79 | 0.43 | 83.18 | 0.81 | 0.41 |

## C  Further Discussions

### C.1  Interpretation of MI-guided Pruning

We study the Mutual Information between an event $\mathbf{v}_i$ and all events $\mathbf{t}_j \in \mathcal{T}$. The normalization notation $\tilde{\cdot}$ is omitted. This local MI measures the relevance between an image token $\mathbf{v}_i$ and all tokens $\mathbf{t}_j \in \mathcal{T}$ in prompts, which is also formulated as the KL-divergence from the marginal distribution $p(\mathcal{T})$ to the

Table 12: **POPE performance comparison under 3 settings on Qwen3VL.** We report accuracy (Acc), F1 scores and Yes-rate.

| Methods | *Random* | | | *Popular* | | | *Adversarial* | | | ***Average*** | | |
|---|---|---|---|---|---|---|---|---|---|---|---|---|
| | Acc ↑ | F1 ↑ | Yes | Acc ↑ | F1 ↑ | Yes | Acc ↑ | F1 ↑ | Yes | Acc ↑ | F1 ↑ | Yes |
| *Qwen3VL-2B* | 92.37 | 0.92 | 0.44 | 89.53 | 0.89 | 0.47 | 87.77 | 0.88 | 0.49 | 89.89 | 0.90 | 0.47 |
| | | | | | | *keep 50%* | | | | | | |
| Random | 91.67 | 0.91 | 0.44 | 88.47 | 0.88 | 0.46 | 86.13 | 0.86 | 0.48 | 88.76 | 0.88 | 0.46 |
| Attention | 92.33 | 0.92 | 0.44 | 89.40 | 0.89 | 0.47 | 86.90 | 0.87 | 0.50 | 89.54 | 0.89 | 0.47 |
| Similarity | 92.63 | 0.92 | 0.45 | 88.77 | 0.89 | 0.49 | 86.37 | 0.87 | 0.51 | 89.26 | 0.89 | 0.48 |
| **MI-Pruner** | 92.94 | 0.93 | 0.45 | 89.33 | 0.89 | 0.48 | 87.07 | 0.87 | 0.50 | 89.78 | 0.90 | 0.48 |
| **MI-Attention** | 92.77 | 0.92 | 0.45 | 89.40 | 0.89 | 0.48 | 87.03 | 0.87 | 0.50 | 89.73 | 0.89 | 0.48 |
| | | | | | | *keep 25%* | | | | | | |
| Random | 88.10 | 0.87 | 0.40 | 84.97 | 0.84 | 0.42 | 83.70 | 0.83 | 0.45 | 85.59 | 0.85 | 0.42 |
| Attention | 88.20 | 0.87 | 0.40 | 86.43 | 0.85 | 0.41 | 84.53 | 0.83 | 0.43 | 86.39 | 0.85 | 0.41 |
| Similarity | 91.33 | 0.91 | 0.44 | 88.13 | 0.88 | 0.47 | 85.37 | 0.85 | 0.49 | 88.28 | 0.88 | 0.47 |
| **MI-Pruner** | 92.73 | 0.92 | 0.45 | 89.13 | 0.89 | 0.48 | 86.67 | 0.87 | 0.51 | 89.51 | 0.89 | 0.48 |
| **MI-Attention** | 92.07 | 0.92 | 0.4 | 88.93 | 0.89 | 0.47 | 87.13 | 0.87 | 0.50 | 89.38 | 0.89 | 0.46 |
| *Qwen3VL-8B* | 90.77 | 0.90 | 0.42 | 88.67 | 0.88 | 0.44 | 87.07 | 0.87 | 0.46 | 88.84 | 0.88 | 0.44 |
| | | | | | | *keep 50%* | | | | | | |
| Random | 89.93 | 0.89 | 0.41 | 88.20 | 0.87 | 0.44 | 86.63 | 0.86 | 0.45 | 88.25 | 0.87 | 0.43 |
| Attention | 90.70 | 0.90 | 0.42 | 88.73 | 0.88 | 0.44 | 87.13 | 0.87 | 0.46 | 88.85 | 0.88 | 0.44 |
| Similarity | 91.07 | 0.90 | 0.43 | 88.67 | 0.88 | 0.45 | 87.10 | 0.87 | 0.47 | 88.95 | 0.88 | 0.45 |
| **MI-Pruner** | 91.43 | 0.91 | 0.45 | 89.00 | 0.89 | 0.43 | 87.23 | 0.87 | 0.47 | 89.22 | 0.89 | 0.45 |
| **MI-Attention** | 90.97 | 0.90 | 0.42 | 88.93 | 0.88 | 0.45 | 87.10 | 0.87 | 0.46 | 89.00 | 0.88 | 0.44 |
| | | | | | | *keep 25%* | | | | | | |
| Random | 87.20 | 0.86 | 0.39 | 85.10 | 0.84 | 0.41 | 84.63 | 0.83 | 0.42 | 85.64 | 0.84 | 0.41 |
| Attention | 89.60 | 0.89 | 0.41 | 87.70 | 0.87 | 0.43 | 85.93 | 0.85 | 0.44 | 87.74 | 0.87 | 0.43 |
| Similarity | 90.43 | 0.90 | 0.43 | 87.40 | 0.87 | 0.46 | 86.10 | 0.86 | 0.47 | 87.98 | 0.88 | 0.45 |
| **MI-Pruner** | 90.70 | 0.90 | 0.43 | 88.77 | 0.88 | 0.45 | 87.00 | 0.87 | 0.46 | 88.82 | 0.88 | 0.45 |
| **MI-Attention** | 90.93 | 0.90 | 0.42 | 88.55 | 0.88 | 0.45 | 87.00 | 0.86 | 0.46 | 88.82 | 0.88 | 0.44 |

conditional distribution $p(\mathcal{T}|\mathbf{v}_i)$:

$$\mathrm{MI}(\mathcal{T}; \mathbf{v}_i) = \sum_{j=1}^{N_T} p(\mathbf{t}_j|\mathbf{v}_i) \log \frac{p(\mathbf{t}_j|\mathbf{v}_i)}{p(\mathbf{t}_j)} \tag{55}$$

$$= D_{\mathrm{KL}}(p(\mathcal{T}|\mathbf{v}_i)\|p(\mathcal{T})) \tag{56}$$

However, estimating these two probabilities $p(\mathbf{t}_j|\mathbf{v}_i)$ and $p(\mathbf{t}_j)$ is non-trivial in high-dimensional projection spaces. Previous approaches often resort to the matrix rank, determinant or kernel-based diversity measures (Zhang et al., 2025d), which suffer from computational expense for matrix inversion or decomposition, and are highly sensitive to numerical instability. In contrast, we adopt the cosine-based Boltzmann distribution with a softmax operation, maintaining a balance between theoretical soundness and computational tractability.

## C.2 Pruning Stages

The data processing inequality (Jaynes, 1957), indicates "processing cannot increase information". Building on this principle, we perform pruning in the projection space, where visual and textual features are semantically aligned but have not yet undergone crossmodal interaction, thereby minimizing the risk of introducing noisy dependencies. Our motivation is similar to the Q-Former (Li et al., 2023), while theoretically grounded without extra training. It's also possible to conduct MI-guided pruning merely on the image features, *i.e.*, after the vision encoder like VisPruner (Zhang et al., 2025c). However, the prompt-agnostic pruning falls short of a truly multimodal setting, since it overlooks the text information.

### C.3   Pruning Levels

Model pruning can be applied at different levels, including weights, architecture and tokens (features). The weight compression tends to be connected with special hardware for acceleration, and the architecture pruning includes layer-level and head-level. Similar to our work, Fan et al. (2021) leverages Mutual Information to prune layers in a top-to-bottom manner. Recent work (Voita et al., 2019; Wang et al., 2021; Kang et al., 2025b) points out that only a few attention heads are necessary in transformer blocks.

### C.4   Settings and Influences

Following existing benchmarks (Zhang et al., 2025f; Chen et al., 2024a), we test our method under a given token budget (*e.g.* $N_{\text{keep}} = 50\%$ means keeping 50% visual tokens). However, a more practical and challenging setup would be, *given a minimum performance threshold and a maximum computation limit, the pruning algorithm automatically decides the trade-off between accuracy and efficiency.* We consider this "dynamic budget" setting as our future work, *i.e.*, to determine the optimal number of tokens for each input adaptively. In addition, pruning holds the potential to mitigate hallucination (Che et al., 2025). Our experiments on LLaVA1.5 (Liu et al., 2024a) demonstrate that heavy pruning on POPE (Li et al., 2025a) doesn't degrade performance compared to the full-budget setting, and on Qwen3-VL (Bai et al., 2025), **MI-Pruner** even leads to improved POPE performance. We attribute these gains to the reduced reasoning difficulty, where less visual information attends to LLM decoding. However, inappropriate pruning can exacerbate hallucinations, as illustrated in Fig. 7. These findings suggest that strategic token pruning represents a promising direction for mitigating hallucination in MLLMs.

## D   Datasets

**GQA**   (Hudson & Manning, 2019). The GQA benchmark is designed to evaluate structured spatial understanding and reasoning within visual scenes. Beyond images and questions, it provides comprehensive scene graph annotations, offering structured representations of objects, attributes, and their inter-relationships. For evaluation, we report the accuracy on the test-dev split, which comprises 12,578 image-question pairs.

**SQA**   (Lu et al., 2022). The ScienceQA benchmark employs multiple-choice questions to assess a model's performance in the scientific domains. The dataset spans three primary subjects—natural science, language science, and social science—and features a hierarchical structure organized by topic, category, and skill. This hierarchy comprises 26 topics, 127 categories, and 379 skills. While the questions are accompanied by relevant illustrations, a portion of the dataset is text-only. For our evaluation, we focus on the SQA-IMG subset, which consists of 2,017 multimodal pairs where both images and questions are present.

**VQA**$_{\text{text}}$   (Singh et al., 2019). The VQA$_{\text{text}}$ benchmark works to assess a model's proficiency in reading and reasoning over visual text. It emphasizes the integration of Optical Character Recognition (OCR) with natural language understanding. The images, primarily sourced from OpenImages-v3 (Krasin et al., 2017), feature diverse real-world scenarios—such as street signs, billboards, and product packaging—that are rich in textual content. Alongside the raw images, ground-truth OCR tokens are provided as auxiliary input. To arrive at the correct answer, models must either extract text directly from the image or perform contextual reasoning based on the identified characters. We report evaluation results on the validation set, which comprises 5,000 image-question pairs.

**MMVet**   (Yu et al., 2023). The MM-Vet benchmark includes 6 core capabilities, including recognition, OCR, knowledge, language generation, spatial awareness, and mathematics, which are combined into 16 specific tasks. Instead of given annotations, this benchmark utilizes GPT-4.1 (OpenAI, 2024) to evaluate its 218 image-question pairs.

**MME**$_{\text{P}}$   (Fu et al., 2025). The MME benchmark encompasses 14 subtasks, including perception and cognition categories. We focus on the perception part (MME$_{\text{P}}$), which includes OCR, coarse-grained recognition

(presence, count, position and color) and fine-grained recognition (posters, celebrities, scenes, landmarks and artworks). All of the 2,374 questions belong to binary judgment tasks.

**POPE** (Li et al., 2025a). The POPE benchmark evaluates the object hallucination in large vision-language models with "Yes-or-No" questions. The images are from MSCOCO dataset (Lin et al., 2014), and the questions are about whether a specific object is present in the image. We report the average Accuracy, F1 score and Yes-rate across three different sampling strategies in the main paper. Notably, we use the latest version, where three strategies include random (3,000), popular (3,000) and adversarial (3,000), leading to overall 9,000 image-question pairs.

**Video datasets.** Video benchmarks extend the image-based VQA into video domain. The GIFs in TGIF-QA (Jang et al., 2017) are based on Tumblr GIF dataset (Li et al., 2016), while MSVD-QA (Xu et al., 2017), MSRVTT-QA (Xu et al., 2017) incorporate Microsoft Research Video Description Corpus (Chen & Dolan, 2011) and Microsoft Research Video to Text (Xu et al., 2016) dataset respectively. Following previous work (Yang et al., 2025a), we test on the first 1,000 samples of all three datasets. All of them are scored by GPT-3.5-Turbo (OpenAI, 2023).

