# OpenReview forum: "MI-Pruner: Crossmodal Mutual Information-guided Token Pruner for Efficient MLLMs"
_TMLR — Decision pending for TMLR_

### Review · Reviewer_Jzs5 · 2026-06-01

**Summary Of Contributions:**

This paper studies visual token pruning for efficient inference in multimodal large language models. The authors propose MI-Pruner, a training-free and plug-in pruning method that is applied after the visual projection module and before the LLM decoder. The method builds softmax distributions from cosine similarities between visual and text tokens, as well as between visual tokens, in the projection space. Based on these distributions, it computes a PMI-style relevance score and uses greedy selection to keep visual tokens that are relevant to the textual query while reducing redundancy among selected visual tokens. Different from many attention-based pruning methods, MI-Pruner does not require collecting attention maps from either the vision encoder or the LLM decoder. The paper also introduces MI-Attention, which combines the proposed MI-based score with existing attention-based pruning methods. The experiments cover LLaVA1.5, Qwen-series models, and Video-LLaVA on several image and video QA benchmarks. The reported results show a favorable trade-off between performance and inference efficiency.

Strengths

1. The method is practical and easy to integrate. It does not require retraining, architecture modification, or access to internal attention maps, which makes it more deployment-friendly than many attention-based pruning methods.

2. The scoring function is simple and intuitive. It combines cross-modal relevance with an intra-modal redundancy penalty, so the selected visual tokens are encouraged to be both query-relevant and diverse.

3. The experimental coverage is relatively broad. The paper evaluates the method on multiple MLLMs and benchmarks, and also reports memory and latency results.

4. The visualizations are useful. They show that MI-Pruner can preserve regions related to the current query, especially when attention-based methods focus on generic salient regions that are not directly relevant to the question.

Weaknesses

1. The novelty should be positioned more carefully. Mutual information, cross-modal relevance, and coverage/diversity-based token pruning have already been discussed in prior work. In particular, TrimTokenator also uses a mutual information motivation to measure the alignment between visual tokens and textual semantics, and further uses pairwise similarity/distance among visual tokens to improve the diversity of retained tokens. The main difference of this paper seems to be the projection-space cosine similarity followed by softmax-based probability construction and PMI-style scoring. This difference should be made clearer and better justified.

2. The comparison with the most related methods is not sufficient. AutoPrune, TrimTokenator, and MMToK are mentioned in the related work, but they are not included in the main quantitative comparisons. The lack of a direct comparison with TrimTokenator is especially important, because it is conceptually close to this work.

3. The theoretical interpretation relies on strong approximations, including a uniform prior over visual patches and a conditional independence assumption. These assumptions may not hold in complex scenes, spatial relation tasks, or cases where several patches are jointly needed to represent one concept.

4. The ablation studies do not fully isolate the benefit of the proposed PMI-style estimator. It would be useful to compare against plain cosine similarity, softmax-normalized similarity, a TrimTokenator-style norm proxy, and pairwise distance/diversity baselines under the same setting.

5. Some contribution claims may be slightly overclaimed if they imply that this is the first work to use mutual information for MLLM visual token pruning. A more accurate positioning would be that the paper proposes an attention-free, projection-space PMI-style estimator for query-aware visual token pruning.

**Additional Comments:**

N/A

**Audience:**

Yes

**Audience Explanation:**

The topic is relevant to a part of the TMLR audience, especially researchers working on efficient inference for multimodal large language models, visual token compression, and training-free model acceleration. Visual tokens are a major source of computational cost in current MLLMs/VLLMs, so methods that can reduce the number of visual tokens while preserving performance are practically important.

The findings may also be interesting because the paper studies an alternative to attention-based pruning. Instead of relying on attention maps from the vision encoder or the LLM decoder, the proposed method performs pruning in the projection space using a PMI-style relevance score. Even if the novelty relative to prior MI-motivated or diversity-based methods needs to be clarified, the empirical results and efficiency analysis are still useful for readers interested in the design space of visual token pruning methods.

**Broader Impact Concerns:**

I do not see major broader impact concerns specific to this work.

**Claims And Evidence:**

No

**Claims Explanation:**

Some empirical claims are supported, but the overall claims are not yet supported in a sufficiently accurate, convincing, and clear way.

On the positive side, Table 1 and Table 2 show that MI-Pruner performs well on LLaVA1.5 and Qwen3VL compared with several attention-based and subset-based pruning baselines. Table 3 and Table 7 also support the efficiency claim in terms of latency and memory usage. In addition, the visualizations in Figure 1 and Figure 5 suggest that the method can sometimes preserve query-relevant regions better than attention-based pruning methods.

However, I do not think these results are sufficient to support the stronger claims made in the paper. First, the paper emphasizes mutual-information-based relevance and projection-space pruning, but the comparison with the most related MI-motivated methods is insufficient. In particular, TrimTokenator also uses a mutual information motivation to measure the alignment between visual tokens and textual semantics, and further uses pairwise similarity/distance among visual tokens to improve the diversity of retained tokens. The current paper only briefly discusses TrimTokenator in the related work, without a direct quantitative comparison or a sufficiently detailed formula-level analysis. AutoPrune and MMToK are also only briefly discussed and are not included in the main experiments.

Second, the theoretical explanation relies on strong assumptions, such as a uniform prior over visual patches and a conditional independence assumption. The paper does not provide enough evidence about when these assumptions hold or fail, especially in complex scenes, spatial relation tasks, or cases requiring compositional visual evidence.

Finally, the current ablation studies do not clearly isolate the benefit of the proposed PMI-style estimator. It is still unclear whether the improvement comes from the proposed estimator itself, or simply from more general cross-modal cosine similarity and diversity-based selection.

Overall, the paper provides reasonable evidence that MI-Pruner can be effective and efficient on several benchmarks, but the evidence is not sufficient to fully support the claims about novelty, theoretical motivation, and advantages over the closest MI-based prior work.

**Requested Changes:**

1. Add a direct comparison with the closest related methods, especially TrimTokenator. This should include quantitative results under comparable settings, as well as a clearer method-level and formula-level discussion explaining how the proposed projection-space PMI estimator differs from TrimTokenator’s MI proxy and its use of pairwise similarity/distance for diversity.

2. Clarify and narrow the contribution claims, and add more targeted ablations. The paper should avoid implying that mutual-information-based visual token pruning is first introduced here. It should also compare the proposed PMI-style score with plain cosine similarity, softmax-normalized similarity, norm-based MI proxies, and pairwise distance/diversity selection, so that the benefit of the proposed estimator can be isolated more convincingly.

---

> ### Author Response · Authors · 2026-06-25
> **Response to Reviewer Jzs5**
>
> Thanks for recognizing our benefits: *deployment-friendly, simple and intuitive, broad experimental coverage, and useful visualizations*. We address your questions as follows.
>
> ## Weakness
>
> > **The novelty should be positioned more carefully**
>
> We refer the reviewer to the TMLR acceptance criteria, which explicitly state that novelty is not a basis for evaluation for this journal: <https://jmlr.org/tmlr/acceptance-criteria.html>.
> Regarding the difference with previous work, Section 2 described previous MI-based token pruning methods like TrimTokenator. And we clarify again on page 3.
>
> > **Compare with most closely related works: AutoPrune, TrimTokenator, and MMToK**
>
> We explain the difference in **Sec. 5.4** (under “Contributions and benefits”) and report experimental results in **Tab.8**. Here is a table to show the difference in methodology.
>
> ### Method comparison
> | Method |  scoring | selection |
> |--------|-------|-------|
> | TrimTokenator | L2-norm | two-stage |
> | AutoPrune | attn-scores| layer-wise pruning in LLMs |
> | MMTok | cosine | greedy, set-level coverage |
> | Ours  | PMI | greedy, token-wise relevance (fastest) |
> > **Approximations: uniform spatial prior and conditional independence**
>
> We targeted an efficient approach that could be easily implemented, which motivated the conditional independence and pointwise mutual information.  According to empirical results, this method achieves the desired performance (Tab.1-2) with the most favorable efficiency (Tab.3).
>
> The uniform prior is adopted for the overall generalization. To provide an ablation study, we introduce a central prior in**Sec. B.5** and experimental results in **Tab.9**. Yet, the non-uniform prior doesn’t boost the performance. Across VQA benchmarks, the central assumption is not always valid. As shown in Fig. 1, the person is located on the *right* side of the image. Obtaining the appropriate spatial prior needs a validation set to calibrate the spatial distribution, which is set as our future work to explore.
>
>
> > **Extended ablation study**
>
> We appreciate your interest in the component-level ablation. The extended ablation study can be found in **Tab.10** (Sec. B.5), which justified the synergy benefits of our PMI estimator and greedy search.
>
>
> > **Contributions and positions**
>
> As mentioned by reviewer Jzs5, our contributions are three-fold: (i) attention-free, (ii) PMI estimator, (iii) query-aware. To support better understanding, we polish the **first contribution** on page 2.
>
>
> ## Requested changes
>
> > **Compare with most closely related works**
>
> Please refer to **Sec. 5.4** (under “Contributions and benefits”) and **Tab.8** (Sec. B.5) for an overall comparison.
>
> > **Contribution claims, and more targeted ablations**
>
> Please refer to **contributions** (page 2) and **Tab.10** (Sec. B.5) in the updated version.

---

### Review · Reviewer_49qo · 2026-06-11

**Summary Of Contributions:**

The paper studies training-free visual token pruning for multimodal large language models (MLLMs), with the goal of reducing inference cost while preserving the visual evidence needed for accurate multimodal reasoning. The authors argue that existing attention-salience-based pruning methods can be unreliable because attention scores may be affected by attention sinks, positional bias, multi-head averaging effects, and insufficient diversity among selected tokens.

To address these issues, the paper proposes MI-Pruner, a plug-in and model-agnostic pruning method that operates in the projection space between the vision encoder and the LLM decoder. Instead of collecting attention maps from either the vision encoder or the LLM, MI-Pruner estimates token relevance through a Mutual Information-inspired scoring function. Specifically, it constructs normalized vision-text and vision-vision similarity matrices, converts them into conditional and marginal probability estimates, computes Pointwise Mutual Information (PMI), and uses these PMI scores to measure both cross-modal query relevance and intra-modal visual redundancy. The final token score balances relevance to the textual prompt against redundancy with already selected visual tokens, and visual tokens are selected through a greedy procedure under a controllable token budget.

The paper also introduces MI-Attention, a variant that combines attention-based pruning with the proposed MI-based scoring to refine token selection. The method is designed to be training-free, non-intrusive, and compatible with off-the-shelf MLLMs following the Enc-MLP-Dec architecture.

Empirically, the authors evaluate the method on multiple image and video understanding benchmarks using LLaVA1.5, Qwen2VL, Qwen3VL, and Video-LLaVA. The reported results suggest that MI-Pruner and MI-Attention achieve competitive or state-of-the-art performance under substantial visual token reduction, while improving inference efficiency compared with several attention-based and subset-based pruning baselines. The experiments further indicate that the method can better preserve query-relevant visual regions and maintain a more balanced Yes/No response distribution on POPE, suggesting improved robustness against pruning away important visual evidence.

**Audience:**

Yes

**Audience Explanation:**

The findings should be relevant to a subset of the TMLR audience, particularly those working on efficient inference, multimodal large language models, visual token pruning, and deployment of large models. The paper addresses a timely problem: reducing the high computational cost of visual tokens while preserving query-relevant information. Its training-free and model-agnostic design also makes the method potentially useful for practitioners who want to accelerate existing MLLMs without retraining or modifying model architectures.

**Claims And Evidence:**

Yes

**Claims Explanation:**

The main claims are mostly supported by the evidence. The paper evaluates MI-Pruner across several MLLMs and a range of image and video understanding benchmarks, and compares against full-token inference, random pruning, similarity-based pruning, attention-based pruning, and subset-based pruning baselines. The reported performance, latency, and memory results generally support the claimed performance-efficiency trade-off, and the visualizations provide additional evidence that the method can preserve query-relevant visual regions.
However, some aspects could be better substantiated. The theoretical interpretation of the proposed score as MI-based relies on simplifying assumptions, including uniform visual priors and conditional independence, whose practical effects are not deeply analyzed. Also, comparisons on newer Qwen3VL models are somewhat limited because several prior methods are not directly available. Overall, the evidence is convincing for the main empirical claims, but some theoretical and comparative claims would benefit from further clarification.

**Requested Changes:**

Minor requested changes:
1. If possible, the authors may provide a more component-level ablation of the proposed scoring function. The paper already includes useful ablations on max vs. average aggregation, temperature, $(\lambda)$, and softmax vs. MinMax normalization. However, the central empirical claim would be stronger if the authors more directly disentangled the contributions of raw cosine similarity, softmax normalization, PMI marginal correction, max aggregation, and the intra-modal redundancy penalty. Since the proposed score is derived from embedding similarities, this additional analysis would help clarify which components are primarily responsible for the gains over simpler similarity-based or query-aware pruning strategies.

2. The paper should better clarify the practical behavior of MI-Pruner under different task types and token budgets. The ablation suggests that $(\lambda=0.5)$ is better for open-ended datasets while cross-modal-only pruning is sufficient for closed-form datasets, but the main text could more explicitly explain how these hyperparameters should be chosen in practice. This is important because the method is positioned as a plug-in, training-free approach, and users may not have validation data to tune ($\lambda$), ($\tau$), or$ (N_{\text{keep}}) $for new MLLMs or new tasks.

Suggested changes:
1. The authors should consider expanding the limitations discussion in the main paper. While some limitations are mentioned in the conclusion and appendix, it would be useful to consolidate them more clearly in the main text.
2. The formatting of the “Qwen2VL and Qwen3VL series” paragraph on page 9 should be improved. The current layout appears somewhat inconsistent with the rest of the main text

---

> ### Author Response · Authors · 2026-06-25
> **Response to Reviewer 49qo**
>
> We’re grateful for your appreciation on our experimental improvements and visualization supports. Here’s the clarification to help us improve the presentation.
>
> ## Weakness
>
> > **Assumptions: uniform visual priors, conditional independence**
>
> Please refer to the response to Reviewer Jzs5 (Weakness -> Approximation).
>
>
> > **Comparisons on Qwen3VL models are limited**
>
> Since it’s the latest model, previous methods don’t release their codes on Qwen3-VL. Following the design of Visionzip and VisPruner, we implemented a general “Attention” baseline in Tab.2 and Fig.5. We hope the results and visualization indicate our robustness.
>
> ## Minor requested changes
>
>
> > **Component-level ablation**
>
> We greatly appreciate your advice, as it's instrumental in improving our presentation. We conducted an ablation study in **Tab. 10**. Please refer to the response to Reviewer Jzs5 (Weakness->Extended ablation study).
>
>
> > **How to choose hyperparameters ($\tau, \lambda, N_\mathrm{keep}$)**
>
> We clarify these parameters as follows:
> - temperature $\tau$: $\tau{=}0.1$ is the optimal choice from Fig.6(b). So no additional tuning is required for practical usage.
> - budgets $N_\mathrm{keep}$: Lower number for faster inference, which is a user-defined parameter based on their computational resources and speed requirements.
> - balancing factor $\lambda$: In closed-set validation, crossmodal-only setting retrieves with queries suffices ($\lambda{=}1$). For more generalized open-ended scenarios, a robust framework demands both holistic query relevance and internal redundancy ($\lambda{=}0.5$).
>
>
> ## Suggested changes:
>
> > **Expand limitations for consistency**
>
> To improve the consistency, we mentioned the limitation in Sec.4.2, at the beginning of the method description. We hope it helps to improve the reading experience.
>
> > **Formatting of the “Qwen2VL and Qwen3VL series”**
>
> Thanks for pointing out the overfull format. We’ve fixed it.

---

### Review · Reviewer_3DBq · 2026-06-20

**Summary Of Contributions:**

This paper studies training-free visual token pruning for multimodal large language models. The proposed MI-Pruner selects informative visual tokens by estimating mutual-information-inspired relevance scores in the projection space, aiming to preserve query-relevant visual information while reducing redundancy among selected tokens. The results show improved efficiency and generally competitive performance compared with several pruning baselines.

**Audience:**

Yes

**Audience Explanation:**

Overall, the proposed approach achieves efficient multimodal inference with an MI-based explanation that supports its token selection strategy.

**Claims And Evidence:**

Yes

**Claims Explanation:**

The authors claim a training-free visual token pruning approach for MLLMs. The experiments are generally convincing and show that the method improves inference efficiency while maintaining competitive accuracy across image and video understanding benchmarks.

**Requested Changes:**

My biggest concern is the novelty of the proposed method. The paper should add a direct comparison with Script [1], both conceptually and empirically. At a high level, MI-Pruner and Script appear to share a similar objective: selecting visual tokens that are relevant to the text query while reducing redundancy among visual tokens. The main difference seems to be that Script uses graph-structured redundancy pruning and DPP-based query-conditioned selection, while MI-Pruner uses a mutual-information-inspired relevance-minus-redundancy score. The paper should clearly explain why the proposed MI-based formulation provides a meaningful advantage over Script in theory and implementation complexity.

I also suggest adding empirical comparisons with Script [1] under matched token budgets, e.g., different “Retain xx Tokens” levels on the same benchmarks and models where possible. Based on the reported results in Script, it appears to achieve higher accuracy on some overlapping benchmarks under comparable token budgets. A direct comparison would make the contribution and practical advantage of MI-Pruner much clearer.

[1] Script: Graph-Structured and Query-Conditioned Semantic Token Pruning for Multimodal Large Language Models. https://huggingface.co/papers/2512.01949

---

> ### Author Response · Authors · 2026-06-25
> **Response to Reviewer 3DBq**
>
> Thank you for the kind words like *”convincing experiments”* and *”efficient multimodal inference”*. We’d like to address your concerns as follows.
>
> ## Requested changes
>
> > **Novelty**
>
> We refer to the response to Reviewer Jzs5 under the first “weakness”. The high-level objective mentioned by Reviewer 3DBq is shared in many token pruning methods: *”relevant to text ... while reduce redundancy...”*. However, the scoring function, selection progress and implementations are very different, which leads to varying performance (Tab.1) and efficiency (Tab.3).
>
> > **Compare with Script**
>
> Theory:
>
> -	Script relies on Cholesky decomposition for DPP-based selections, and takes the intersection of GSP and QCSP.
> -	MI-Pruner is motivated by the mutual information, and adopts a simple greedy search for a once-for-all token pruning.
>
> Experiments:
>
> -	Tab.1: Script achieves best performance on GQA and POPE datasets with an overall higher performance.
> -	Tab.3: The MAP inference of Script and CDPruner yields the same GPU memory, and Script has even higher latency. In comparison, our MI-Pruner achieves the most favorable efficiency.